# IMPROVED TRAINING OF CERTIFIABLY ROBUST MODELS

## ABSTRACT

Convex relaxations are effective for training and certifying neural networks against norm-bounded adversarial attacks, but they leave a large gap between certifiable and empirical robustness. In principle, convex relaxation can provide tight bounds if the solution to the relaxed problem is feasible for the original non-convex problem. Therefore, we propose two regularizers that can be used to train neural networks that yield tighter convex relaxation bounds for robustness. In all of our experiments, the proposed regularizers result in higher certified accuracy than non-regularized baselines.

## 1 INTRODUCTION

Neural networks have achieved excellent performances on many computer vision tasks, but they could be vulnerable to small, adversarially chosen perturbations that are barely perceptible to humans while having a catastrophic impact on the network's performance (Szegedy et al., 2013; Goodfellow et al., 2014). Making classifiers robust to these adversarial perturbations is of great interest, especially when neural networks are applied to safety-critical applications. Several heuristic methods exist for obtaining robust classifiers, however powerful adversarial examples can be found against most of these defenses (Carlini & Wagner, 2017; Uesato et al., 2018).

Recent studies focus on verifying or enforcing the certified accuracy of deep classifiers, especially for networks with ReLU activations. They provide guarantees of a network's robustness to any perturbation $\delta$ with norm bounded by $\|\delta\|_p \leq \epsilon$ (Wong & Kolter, 2017; Wong et al., 2018; Raghunathan et al., 2018; Dvijotham et al., 2018b; Zhang et al., 2018; Salman et al., 2019). There are exact verifiers that find the exact minimum adversarial distortions $\delta$ or the robust error (Ehlers, 2017; Katz et al., 2017; Tjeng et al., 2017), but due to the non-convex nature of the problem, exact verification is NP-hard. To make verification efficient and scalable, convex relaxations are adopted to expand the non-convex feasible set given by non-linear activations into convex sets, resulting in a lower bound on the norm of adversarial perturbations (Zhang et al., 2018; Weng et al., 2018), or an upper bound on the robust error (Dvijotham et al., 2018b; Gehr et al., 2018; Singh et al., 2018). With LP relaxations (Wong et al., 2018), such a verifier can be efficient enough to estimate the lower bound of the margin in each iteration for training certifiably robust networks. However, due to the relaxation of the underlying problem, the lower bound of the margin is potentially loose, and thus a barrier remains between the optimal values from the original and relaxed problems (Salman et al., 2019).

In this paper, we focus on improving the certified robustness of neural networks trained with convex relaxation bounds. To achieve this, we first give a more interpretable explanation for the bounds achieved in (Weng et al., 2018; Wong et al., 2018). Namely, the constraints of the relaxed problem are defined by a simple linear network with adversaries injecting bounded perturbations to both the input of the network and the pre-activations of intermediate layers. The optimal solution of the relaxed problem can be written as a forward pass of the clean image through the linear network, plus the cumulative adversarial effects of all the perturbations added to the linear transforms, which makes it easier to identify the optimality conditions and serves as a bridge between the relaxed problem and the original non-convex problem. We further identify conditions for the bound to be tight, and we propose two indicators for the gap between the original non-convex problem and the relaxed problem. Adding the proposed indicators into the loss function results in classifiers with better certified accuracy.

## 2    BACKGROUND AND RELATED WORK

Adversarial defenses roughly fall into two categories: heuristic defenses and verifiable defenses. The heuristic defenses either try to identify adversarial examples and remove adversarial perturbations from images, or make the network invariant to small perturbations through training (Papernot & McDaniel, 2018; Shan et al., 2019; Samangouei et al., 2018; Hwang et al., 2019). In addition, adversarial training uses adversarial examples as opposed to clean examples during training, so that the network can learn how to classify adversarial examples directly (Madry et al., 2017; Shafahi et al., 2019; Zhang et al., 2019a).

In response, a line of works have proposed to verify the robustness of neural nets. Exact methods obtain the perturbation $\delta$ with minimum $\|\delta\|_p$ such that $f(x) \neq f(x + \delta)$, where $f$ is a classifier and $x$ is the data point. Nevertheless, the problem itself is NP-hard and the methods can hardly scale (Cheng et al., 2017; Lomuscio & Maganti, 2017; Dutta et al., 2018; Fischetti & Jo, 2017; Tjeng et al., 2017; Scheibler et al., 2015; Katz et al., 2017; Carlini et al., 2017; Ehlers, 2017).

A body of work focuses on relaxing the non-linearities in the original problem into linear inequality constraints (Singh et al., 2018; Gehr et al., 2018; Zhang et al., 2018; Mirman et al., 2018), sometimes using the dual of the relaxed problem (Wong & Kolter, 2017; Wong et al., 2018; Dvijotham et al., 2018b). Recently, Salman et al. (2019) unified the primal and dual views into a common convex relaxation framework, and suggested there is an inherent gap between the actual and the lower bound of robustness given by verifiers based on LP relaxations, which they called a *convex relaxation barrier*.

Some defense approaches integrate the verification methods into the training of a network to minimize robust loss directly. Hein & Andriushchenko (2017) uses a local lipschitz regularization to improve certified robustness. In addition, a bound based on semi-definite programming (SDP) relaxation was developed and minimized as the objective (Raghunathan et al., 2018). Wong & Kolter (2017) presents an upper bound on the robust loss caused by norm-bounded perturbation via LP relaxation, and minimizes this upper bound during training. Wong et al. (2018) further extend this method to much more general network structures with skip connections and general non-linearities, and provide a memory-friendly training strategy using random projections. Since LP relaxation is adopted, the aforementioned convex relaxation barrier exists for their methods.

While another line of work (IBP) have shown that an intuitively looser interval bound can be used to train much more robust networks than convex relaxation for large $\ell_\infty$ perturbations (Gowal et al., 2018; Zhang et al., 2019b), it is still important to study convex relaxation bounds since it can provide better certificates against a broader class of adversaries that IBP struggles to certify in some cases, such as $\ell_2$ adversaries for convolutional networks. We give a brief discussion for the reason in Appendix E.

We seek to enforce the tightness of the convex relaxation certificate during training. It reduces the optimality gap between the original and the relaxed problem by adding the proposed indicators into the loss function as regularizers. Compared with previous approaches, we have the following contributions: First, based upon the same relaxation in (Weng et al., 2018), we illustrate a more intuitive view for the bounds on intermediate ReLU activations achieved by (Wong et al., 2018) , which can be viewed as a linear network facing adversaries adding bounded perturbations to both the input and the intermediate layers. Second, starting from this view, we identify conditions where the bound from the relaxed problem is tight for the original non-convex problem. Third, based on the conditions, we propose regularizers that encourage the bound to be tight for the obtained network, which improves the certificate on both MNIST and CIFAR-10.

## 3    PROBLEM FORMULATION

In general, to train an adversarially robust network, we solve a constrained minimax problem where the adversary tries to maximize the loss given the norm constraint, and the parameters of the network are trained to minimize this maximal loss. Due to nonconvexity and the complexity of neural networks, it is expensive to solve the inner max problem exactly. To obtain certified robustness, like many related works (Wong et al., 2018; Gowal et al., 2018), we minimize an upper bound of the inner max problem, which is a cross entropy loss on the negation of the lower bounds of margins over each

other class, as shown in Eq. 4. Without loss of generality, in this section we analyze the original and relaxed problems for minimizing the margin between the ground truth class $y$ and some other class $t$ under norm-bounded adversaries, which can be adapted directly to compute the loss in Eq. 4.

The original nonconvex constrained optimization problem for finding the norm-bounded adversary that minimizes the margin can be formulated as

$$\min_{z_1 \in \mathcal{B}_{p,\epsilon}(x)} c_t^\top x_L, \text{ subject to } z_{i+1} = \sigma(x_i), x_i = f_i(z_i), \text{ for } i = 1, ..., L, \qquad (\mathcal{O})$$

where $c_t = e_y - e_t$, $e_y$ and $e_t$ are one-hot vectors corresponding to the label $y$ and some other class $t$, $\sigma(\cdot)$ is the ReLU activation, and $f_i$ is one functional block of the neural network. This can be a linear layer ($f_i(z_i) = W_i z_i + b_i$), or even a residual block. We use $h_i(x) = f_i(\sigma(f_{i-1}(\cdots f_1(x))))$ to denote the ReLU network up to the $i$-th layer, and $p^*_{\mathcal{O}}$ to denote the optimal solution to $\mathcal{O}$.

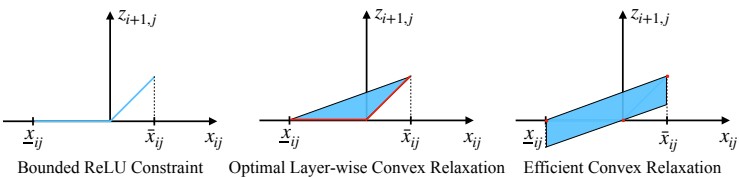

Figure 1: The feasible sets (blue regions/lines) given by the bounded ReLU constraints (Eq. $\mathcal{O}$), convex hull ($\overline{conv}_{ij}$) and the relaxation (Fast-Lin) discussed in this paper (specific choice for Eq. $\mathcal{C}$) for $j \in \mathcal{I}_i$. The red lines and dots are the intersections between the boundaries of the convex feasible sets and the ReLU constraints.

### 3.1 EFFICIENT CONVEX RELAXATIONS

**Grouping of ReLU Activations**  The nonconvexity of $\mathcal{O}$ stems from the nonconvex feasible set given by the ReLU activations. Since the network is a continuous function, the pre-activations $x_i$ have lower and upper bounds $\underline{x}_i$ and $\bar{x}_i$ when the input $z_1 \in \mathcal{B}_{p,\epsilon}(x)$. If a certain pre-activation $x_{ij}$ has $\underline{x}_{ij} < 0 < \bar{x}_{ij}$, its corresponding ReLU constraint $z_{i+1,j} = \sigma(x_{ij})$ gives rise to a non-convex feasible set as shown in the left of Figure 1, making Eq. $\mathcal{O}$ a non-convex optimization problem. On the other hand, if $\bar{x}_{ij} \leq 0$ or $\underline{x}_{ij} \geq 0$, the constraints degenerate into linear constraints $z_{i+1,j} = 0$ and $z_{i+1,j} = x_{ij}$ respectively, which do not affect convexity. Based on $\underline{x}_i$ and $\bar{x}_i$, we divide the ReLU activations into three disjoint subsets

$$\mathcal{I}_i^- = \{j | \bar{x}_{ij} \leq 0\}, \quad \mathcal{I}_i^+ = \{j | \underline{x}_{ij} \geq 0\}, \quad \mathcal{I}_i = \{j | \underline{x}_{ij} < 0 < \bar{x}_{ij}\}. \qquad (1)$$

If $j \in \mathcal{I}_i$, we call the corresponding ReLU activation an *unstable neuron*.

Convex relaxation expands the non-convex feasible sets into convex ones and solves a convex optimization problem $\mathcal{C}$. The feasible set of $\mathcal{O}$ is a subset of the feasible set of $\mathcal{C}$, so the optimal value of $\mathcal{C}$ lower bounds the optimal value of Eq. $\mathcal{O}$. Moreover, we want problem $\mathcal{C}$ to be solved efficiently, better with a closed form solution, so that it can be integrated into the training process.

**Computational Challenge for the "optimal" Relaxation**  As pointed out by (Salman et al., 2019), the optimal layer-wise convex relaxation, i.e., the optimal convex relaxation for the nonlinear constraint $z_{i+1} = \sigma(x_i)$ of a single layer, can be obtained independently for each neuron. For each $j \in \mathcal{I}_i$ in a ReLU network, the optimal layer-wise convex relaxation is the closed convex hull $\overline{conv}_{ij}$ of $\mathcal{S}_{ij} = \{(x_{ij}, z_{i+1,j}) | j \in \mathcal{I}_i, z_{i+1,j} = \max(0, x_{ij}), \underline{x}_{ij} \leq x_{ij} \leq \overline{x}_{ij}\}$, which is just $\overline{conv}_{ij} = \{(x_{ij}, z_{i+1,j}) | \max(0, x_{ij}) \leq z_{i+1,j} \leq \frac{\overline{x}_{ij}}{\overline{x}_{ij} - \underline{x}_{ij}}(x_{ij} - \underline{x}_{ij})\}$, corresponding to the triangle region in the middle of Figure 1. Despite being relatively tight, there is no closed-form solution to this relaxed problem. LP solvers are typically adopted to solve a linear programming problem for each neuron. Therefore, such a relaxation is hardly scalable to verify larger networks without any additional trick (like Xiao et al. (2018)). Weng et al. (2018) find it to be 34 to 1523 times slower than Fast-Lin, and it has difficulty verifying MLPs with more than 3 layers on MNIST. In (Salman et al., 2019), it takes 10,000 CPU cores to parallelize the LP solvers for bounding the activations of every neuron in a two-hidden-layer MLP with 100 neurons per layer. Since solving LP problems for all neurons are usually impractical, it is even more difficult to optimize the network to maximize the

lower bounds of margin found by solving this relaxation problem, as differentiating through the LP optimization process is even more expensive.

**Computationally Efficient Relaxations**   In the layer-wise convex relaxation, instead of using a boundary nonlinear in $x_{ij}$, (Zhang et al., 2018) has shown that for any nonlinearity, when both the lower and upper boundaries are linear in $x_{ij}$, there exist closed-form solutions to the relaxed problem, which avoids using LP solvers and improves efficiency. Specifically, the following relaxation of $\mathcal{O}$ has closed-form solutions:

$$
\begin{aligned}
\text{minimize} \quad & c_t^\top x_L, \\
\text{subject to} \quad & \underline{a}_i \cdot x_i + \underline{b}_i \le z_{i+1} \le \bar{a}_i \cdot x_i + \bar{b}_i, \underline{x}_i \le x_i \le \bar{x}_i, \\
& x_i = W_i z_i + b_i, \text{for } i = 1, ..., L, z_1 \in \mathcal{B}_{p,\epsilon}(x),
\end{aligned} \tag{$\mathcal{C}$}
$$

where $\cdot$ denotes element-wise product, and for simplicity, we have only considered networks with no skip connections, and represent both Full Connected and Convolutional Layers as a linear transform $f_i(z_i) = W_i z_i + b_i$.

Before we can solve $\mathcal{C}$ to get the lower bound of margin, we need to know thee range $[\underline{x}_i, \bar{x}_i]$ for the pre-activations $x_i$. As in (Wong & Kolter, 2017; Weng et al., 2018; Zhang et al., 2018), we can solve the same optimization problem for each neuron $x_{ij}$ starting from layer 1 to $L$, by replacing $c_t$ with $e_j$ or $-e_j$ for $\underline{x}_{ij}$ or $\bar{x}_{ij}$ respectively.[1]

The most efficient approach in this category is Fast-Lin (Weng et al., 2018), which sets $\underline{a}_{ij} = \bar{a}_{ij}$, as shown in the right of Figure 1. A tighter choice is CROWN (Zhang et al., 2018), which chooses different $\underline{a}_{ij}$ and $\bar{a}_{ij}$ such that the convex feasible set is minimized. However, CROWN has much higher complexity than Fast-Lin due to its varying slopes. We give detailed analysis of the closed-form solutions of both bounds and their complexities in Appendix F. Recently, CROWN-IBP (Zhang et al., 2019b) has been proposed to provide a better initialization to IBP, which uses IBP to estimate range $[\underline{x}_i, \bar{x}_i]$ for CROWN. In this case, both CROWN and Fast-Lin have the same complexity and CROWN is a better choice.

# 4   TIGHTER BOUNDS VIA REGULARIZATION

Despite being relatively efficient to compute, Fast-Lin and CROWN are not even the tightest layer-wise convex relaxation. Using tighter bounds to train the networks could potentially lead to higher certified robustness by preventing such bounds from over-regularizing the networks. Nevertheless, there exist certain parameters and inputs such that the seemingly looser Fast-Lin is tight for $\mathcal{O}$, i.e., the optimal value of Fast-Lin is the same as $\mathcal{O}$. The most trivial case is where no unstable neuron exists. In practice, as shown in the illustrative example in Appendix A, Fast-Lin can be tight even when unstable neurons exist. It is therefore interesting to check the conditions for Fast-Lin or CROWN to be tight for $\mathcal{O}$, and enforcing such conditions during training to improve the certified robustness.

## 4.1   CONDITIONS FOR TIGHTNESS

Here we look into conditions that make the optimal value $p_{\mathcal{C}}^*$ of the convex problem $\mathcal{C}$ to be equal to $p_{\mathcal{O}}^*$. Let $\{z_i, x_i\}_{i=1}^L$ be some feasible solution of $\mathcal{C}$, from which the objective value of $\mathcal{C}$ can be determined as $p_\mathcal{C} = c_t^\top x_L$. Let $\{z_i', x_i'\}_{i=1}^L$ be some feasible solution of $\mathcal{O}$ computed by passing $z_1'$ through the ReLU sub-networks $h_i(z_1')$ defined in $\mathcal{O}$, and denote the resulting feasible objective value as $p_{\mathcal{O}}' = c_t^\top x_L'$.

Generally, for a given network with the set of weights $\{W_i, b_i\}_{i=1}^L$, as long as the optimal solution $\{z_i^*, x_i^*\}_{i=1}^L$ of $\mathcal{C}$ is equal to a feasible solution $\{z_i', x_i'\}_{i=1}^L$ of $\mathcal{O}$, we will have $p_{\mathcal{O}}^* = p_{\mathcal{C}}^*$, since any feasible $p_{\mathcal{O}}'$ of $\mathcal{O}$ satisfies $p_{\mathcal{O}}' \ge p_{\mathcal{O}}^*$, and by the nature of relaxation $p_{\mathcal{C}}^* \le p_{\mathcal{O}}^*$.

Therefore, for a given network and input $x$, to check the tightness of the convex relaxation, we can check whether its optimal solution $\{z_i^*, x_i^*\}_{i=1}^L$ is feasible for $\mathcal{O}$. This can be achieved by passing $z_1^*$ through the ReLU network, and either directly check the resultant objective value $p_{\mathcal{O}}'$, or compare $\{z_i^*, x_i^*\}_{i=1}^L$ with the resultant feasible solution $\{z_i', x_i'\}_{i=1}^L$. Further, we can encourage such

---

[1]For $\bar{x}_{ij}$, take an extra negation on the solution.

conditions to happen during the training process to improve the tightness of the bound. Based on such mechanisms, we propose two regularizers to enforce the tightness. Notice such regularizers are different from the RS Loss (Xiao et al., 2018) introduced to reduce the number of unstable neurons, since we have shown with Appendix A that $\mathcal{C}$ can be tight even when unstable neurons exist.

## 4.2 A Intuitive Indicator of Tightness: $p'_{\mathcal{O}} - p^*_{\mathcal{C}}$

The observation above motivates us to consider the non-negative value

$$d(x, \delta^*_0, W, b) = p'_{\mathcal{O}}(x, \delta^*_0) - p^*_{\mathcal{C}} \tag{2}$$

as an indicator of the difference between $\{z^*_i, x^*_i\}^L_{i=1}$ and $\{z'_i, x'_i\}^L_{i=1}$, where $p'_{\mathcal{O}}(x, \delta^*_0) = c^\top_t h_L(x + \delta^*_0)$ is the margin over class $t$ computed by passing the optimal perturbation $\delta^*_0$ for $\mathcal{C}$ through the original network. $\delta^*_0$ can be computed efficiently from the optimality condition of Fast-Lin or CROWN, as demonstrated in Eq. 16. For example, when $p = \infty$, the optimal input perturbation $\delta^*_0$ of $\mathcal{C}$ is $\delta^*_0 = -\epsilon \text{sign}(c^\top_t \mathcal{W}_{L:1})$, which corresponds to sending $z'_1 = z^*_1 = x - \epsilon \text{sign}(c^\top_t \mathcal{W}_{L:1})$ through the ReLU network; when $p = 2$, $\delta^*_0 = -\epsilon \frac{c^\top_t \mathcal{W}_{L:1}}{\|c^\top_t \mathcal{W}_{L:1}\|_2}$, which corresponds to sending $z'_1 = z^*_1 = x - \epsilon \frac{c^\top_t \mathcal{W}_{L:1}}{\|c^\top_t \mathcal{W}_{L:1}\|_2}$.

The larger $d(x, \delta^*_0, W, b)$ is, the more relaxed $\mathcal{C}$ is, and the higher $p^*_{\mathcal{O}} - p^*_{\mathcal{C}}$ could be. Therefore, we can regularize the network to minimize $d(x, \delta^*_0, W, b)$ during training and maximize the lower-bound of the margin $p^*_{\mathcal{C}}$, so that we can obtain a network where $p^*_{\mathcal{C}}$ is a better estimate of $p^*_{\mathcal{O}}$ and the robustness is better represented by $p^*_{\mathcal{C}}$. Such an indicator avoids comparing the intermediate variables, which gives more flexibility for adjustment. It bears some similarities to knowledge distillation (Hinton et al., 2015), in that it encourages learning a network whose relaxed lower bound gives similar outputs of the corresponding ReLU network. It is worth noting that minimizing $d(x, \delta^*_0, W, b)$ does not necessarily lead to decreasing $p'_{\mathcal{O}}(x, \delta^*_0)$ or increasing $p^*_{\mathcal{C}}$. In fact, both $p'_{\mathcal{O}}(x, \delta^*_0)$ and $p^*_{\mathcal{C}}$ can be increased or decreased at the same time with their difference decreasing.

The tightest indicator should give the minimum gap $p^*_{\mathcal{O}} - p^*_{\mathcal{C}}$, where we need to find the optimal perturbation for $\mathcal{O}$. However, the minimum gap cannot be found in polynomial time, due to the non-convex nature of $\mathcal{O}$. (Weng et al., 2018) also proved that there is no polynomial time algorithm to find the minimum $\ell_1$-norm adversarial distortion with $0.99 \ln n$ approximation ratio unless NP=P, a problem equivalent to finding the minimum margin here.

## 4.3 A Better Indicator for Regularization: Difference in Solutions

Despite being intuitive and is able to achieve improvements, Eq. 2 which enforces similarity between objective values does not work as good as enforcing similarity between the solutions $\{z^*_i, x^*_i\}^L_{i=1}$ and $\{z'_i, x'_i\}^L_{i=1}$ in practice, an approach we will elaborate below. For both CROWN and Fast-Lin, unless $d(x, \delta^*_0, W, b) = 0$, $\{z^*_i, x^*_i\}^L_{i=1}$ may deviate a lot from $\{z'_i, x'_i\}^L_{i=1}$ and does not correspond to any ReLU network, even if $d(x, \delta^*_0, W, b)$ may seem small. For example, it is possible that $z^*_{ij} < 0$ for a given $z^*_1$, but a ReLU network will always have $z'_{ij} \geq 0$.

We find an alternative regularizer more effective at improving verifiable accuracy. The regularizer encourages the feasible solution $\{z'_i, x'_i\}^L_{i=1}$ of $\mathcal{O}$ to *exactly* match the feasible optimal solution $\{z^*_i, x^*_i\}^L_{i=1}$ of $\mathcal{C}$. Since we are adopting the layer-wise convex relaxation, the optimal solutions of the unstable neurons can be considered independently.

Here we derive a sufficient condition for tightness for Fast-Lin, which also serves as a sufficient condition for CROWN. For linear programming, the optimal solution occurs on the boundaries of the feasible set. Since Fast-Lin is a layer-wise convex relaxation, the solution to each of its neurons in $z_i$ can be considered independently, and therefore for a specific layer $i$ and $j \in \mathcal{I}_i$, the pair of optimal solutions $(x^*_{ij}, z^*_{i+1,j})$ should occur on the boundary in the right of Figure 1. It follows that the only 3 optimal solutions $(x^*_{ij}, z^*_{i+1,j})$ of $\mathcal{C}$ that are also feasible for $\mathcal{O}$ are $(\underline{x}_{ij}, 0)$, $(\overline{x}_{ij}, \overline{x}_{ij})$ and $(0, 0)$. Notice they are also in the intersection between the boundary of CROWN and $\mathcal{O}$.

In practice, out of efficiency concerns, both Fast-Lin and CROWN identify the boundaries that the optimal solution lies on and computes the optimal value by accumulating the contribution of each layer in a backward pass, without explicitly computing $\{z^*_i, x^*_i\}^L_{i=1}$ for each layer with a forward

pass (see Appendix F for more details). It is therefore beneficial to link the feasible solutions of $\mathcal{O}$ to the parameters of the boundaries. Specifically, let $\delta_{ij}^* \in \{\underline{b}_{ij}, \overline{b}_{ij}\}$ be the intercept of the line that the optimal solution $(x_{ij}^*, z_{i+1,j}^*)$ lies on. We want to find a rule based on $\{\delta_i^*\}_{i=1}^L$ to determine whether the bound is tight from the values of $\{x_i'\}_{i=1}^L$. For both Fast-Lin and CROWN, $\underline{b}_{ij} = 0, \overline{b}_{ij} = -\frac{\overline{x}_{ij}\underline{x}_{ij}}{\overline{x}_{ij}-\underline{x}_{ij}}$. For Fast-Lin, when $\delta_{ij}^* = \overline{b}_{ij}$, only $(x_{ij}^*, z_{i+1,j}^*) = (\underline{x}_{ij}, 0)$ or $(\overline{x}_{ij}, \overline{x}_{ij})$ are fesible for $\mathcal{O}$; when $\delta_{ij}^* = \underline{b}_{ij}$, only $(x_{ij}^*, z_{i+1,j}^*) = (0, 0)$ is feasible for $\mathcal{O}$. Meanwhile, $z_{i+1,j}' = \max(x_{ij}', 0)$ is deterministic if $x_{ij}'$ is given. Therefore, when the bound is tight for Fast-Lin, if $\delta_{ij}^* = \underline{b}_{ij}$, then $x_{ij}' = 0$. Otherwise, if $\delta_{ij}^* = \overline{b}_{ij}$, and $x_{ij}' = \underline{x}_{ij}$ or $\overline{x}_{ij}$. For CROWN, this condition is also feasible, though it could be either $x_{ij}' \leq 0$ or $x_{ij}' \geq 0$ when $\delta_{ij}^* = 0$, depending on the optimal slope $\underline{D}_{ij}^{(L)}$.

Indeed, we achieve optimal tightness ($p_\mathcal{C}^* = p_\mathcal{O}^*$) for both Fast-Lin and CROWN if $x_{ij}'$ satisfy these conditions at *all* unstable neurons. Specifically,

**Proposition 1.** *Assume $\{z_i', x_i'\}_{i=1}^L$ is obtained by the ReLU network $h_L$ with input $z_1'$, and $\{\delta_i^*\}_{i=0}^{L-1}$ is the optimal solution of Fast-Lin or CROWN. If $z_1' = x + \delta_0^*$, and $x_{ij}' \in \mathcal{S}(\delta_{ij}^*)$ for all $i = 1, ..., L-1, j \in \mathcal{I}_i$, then $\{z_i', x_i'\}_{i=1}^L$ is an optimal solution of $\mathcal{O}$, Fast-Lin and CROWN. Here*

$$\mathcal{S}(\delta_{ij}^*) = \begin{cases} \{\underline{x}_{ij}, \overline{x}_{ij}\} & \text{if } \delta_{ij}^* = -\frac{\overline{x}_{ij}\underline{x}_{ij}}{\overline{x}_{ij}-\underline{x}_{ij}}, \\ \{0\} & \text{if } \delta_{ij}^* = 0. \end{cases}$$

We provide the proof of this simple proposition in the Appendix.

It remains to be discussed how to best enforce the similarity between the optimal solutions of $\mathcal{O}$ and Fast-Lin or CROWN. Like before, we choose to enforce the similarity between $\{x_i'\}_{i=1}^L$ and the closest optimal solution of Fast-Lin, where $\{x_i'\}_{i=1}^L$ is constructed by setting $x_1' = x_1^* = W_1(x + \delta_0^*) + b_1$ and pass $x_1'$ through the ReLU network to obtain $x_i' = h_i(x + \delta_0^*)$. By Proposition 1, the distance can be computed by considering the values of the intercepts $\{\delta_i^*\}_{i=1}^{L-1}$ as

$$r(x, \delta_0^*, W, b) = \frac{1}{\sum_{i=1}^{L-1} |\mathcal{I}_i|} \sum_{i=1}^{L-1} \left( \sum_{\substack{j \in \mathcal{I}_i \\ \delta_{ij}^* = 0}} |x_{ij}'| + \sum_{\substack{j \in \mathcal{I}_i \\ \delta_{ij}^* \neq 0}} \min(|x_{ij}' - \underline{x}_{ij}|, |x_{ij}' - \overline{x}_{ij}|) \right), \quad (3)$$

where the first term corresponds to $\delta_{ij}^* = 0$ and the condition $x_{ij}' \in \{0\}$, and the second term corresponds to $\delta_{ij}^* = -\frac{\overline{x}_{ij}\underline{x}_{ij}}{\overline{x}_{ij}-\underline{x}_{ij}}$ and the condition $x_{ij}' \in \{\underline{x}_{ij}, \overline{x}_{ij}\}$. To minimize the second term, the original ReLU network only needs to be optimized towards the nearest feasible optimal solution. It is easy to see from Proposition 1 that if $r(x, \delta_0^*, W, b) = 0$, then $p_\mathcal{O}^* = p_\mathcal{C}^*$, where $\mathcal{C}$ could be both Fast-Lin or CROWN.

Compared with $d(x, \delta_0^*, W, b)$, $r(x, \delta_0^*, W, b)$ puts more constraints on the parameters $W, b$, since it requires all unstable neurons of the ReLU network to match the optimal solutions of Fast-Lin, instead of only matching the objective values $p_\mathcal{O}'$ and $p_\mathcal{C}^*$. In this way, it provides stronger guidance towards a network whose optimal solution for $\mathcal{O}$ and Fast-Lin or CROWN agree. However, again, this is not equivalent to trying to kill all unstable neurons, since Fast-Lin can be tight even when unstable neurons exist.

---

**Algorithm 1** Computing the Fast-Lin Bounds and Regularizers for $\ell_\infty$ Norm

---

**Data:** Clean images $\{x^{(j)}\}_{j=1}^m$, ReLU network with parameters $W, b$, and maximum perturbation $\epsilon$.
**Result:** Margin Lower Bound $p_\mathcal{C}^*$, Regularizer $d(x, \delta_0^*, W, b)$ and $r(x, \delta_0^*, W, b)$.
  Initialize $\mathcal{W}_{1:1} \leftarrow W_1$, $g_1(x) = W_1 x + b_1$, $\underline{x}_1 = g_1(x) - \epsilon \|\mathcal{W}_{1:1}\|_{1,row}$, $\bar{x}_1 = g_1(x) + \epsilon \|\mathcal{W}_{1:1}\|_{1,row}$
**for** $i=2,\ldots,L\text{-}1$ **do**
  Compute $D_{i-1}$ with Eq. 13, $\mathcal{W}_{i:1} \leftarrow W_i D_{i-1} \mathcal{W}_{i-1}$, $g_i(x) = W_i D_{i-1} g_{i-1}(x) + b_i$
  $\underline{x}_i \leftarrow g_i(x) - \epsilon \|\mathcal{W}_{i:1}\|_{1,row} - \sum_{i'=1}^{i-1} \sum_{j \in I_{i'}} \frac{\underline{x}_{i'j} \bar{x}_{i'j}}{\bar{x}_{i'j} - \underline{x}_{i'j}} \min((\mathcal{W}_{i:i'+1})_{:,j}, 0)$
  $\overline{x}_i \leftarrow g_i(x) + \epsilon \|\mathcal{W}_{i:1}\|_{1,row} + \sum_{i'=1}^{i-1} \sum_{j \in I_{i'}} \frac{\underline{x}_{i'j} \overline{x}_{i'j}}{\overline{x}_{i'j} - \underline{x}_{i'j}} \min(-(\mathcal{W}_{i:i'+1})_{:,j}, 0)$
**end**
  Compute $D_{L-1}$ with Eq. 13, $\mathcal{W}_{L:1} \leftarrow W_L D_{L-1} \mathcal{W}_{L-1}$, $g_L(x) = W_L D_{L-1} g_{L-1}(x) + b_L$
  Compute $p_\mathcal{C}^*$ with Eq. 17, $\delta_0^* \leftarrow -\epsilon \|c_t^\top \mathcal{W}_{L:1}\|_{1,row}$
  $x_i' \leftarrow h_i(x + \delta_0^*)$ **for** $i = 1$ to $L$
  $d(x, \delta_0^*, W, b) \leftarrow c_t^\top x_L' - p_\mathcal{C}^*$
  Compute $r(x, \delta_0^*, W, b)$ with Eq. 3

---

## 4.4 Certified Robust Training in Practice

In practice, for classification problems with more than two classes, we will compute the lower bound of the margins w.r.t. multiple classes. Denote $\mathbf{p}_\mathcal{C}^*$ and $\mathbf{p}_\mathcal{O}^*$ as the concatenated vector of lower bounds of the relaxed problem and original problem for multiple classes, and $d_t, r_t$ as the regularizers for the margins w.r.t. class $t$. Together with the regularizers, we optimize the following objective

$$\underset{W,b}{\text{minimize}} \; L_{CE}(-\mathbf{p}_\mathcal{C}^*, y) + \lambda \sum_t d_t(x, \delta_0^*, W, b) + \gamma \sum_t r_t(x, \delta_0^*, W, b), \qquad (4)$$

where $L_{CE}(-\mathbf{p}_\mathcal{C}^*, y)$ is the cross entropy loss with label $y$, as adopted by many related works (Wong et al., 2018; Gowal et al., 2018), and we have implicitly abbreviated the inner maximization problem w.r.t. $\{\delta_i\}_{i=0}^{L-1}$ into the optimal values $\mathbf{p}_\mathcal{C}^*$ and solution $\delta_0^*$. More details for computing the intermediate and output bounds can be found in Algorithm 1, where we have used $\|\cdot\|_{1,row}$ to denote row-wise $\ell_1$ norm, and $(\cdot)_{:,j}$ for taking the $j$-th column.

One major challenge of the convex relaxation approach is the high memory consumption. To compute the bounds $\underline{x}_i, \overline{x}_i$, we need to pass an identity matrix with the same number of diagonal entries as the total dimensions of the input images, which can make the batch size thousands of times larger than usual. To mitigate this, one can adopt the random projection from Wong et al. (2018), which projects identity matrices into lower dimensions as $\mathcal{W}_{i:1} R$ to estimate the norm of $\mathcal{W}_{i:1}$. Such projections add noise/variance to $\underline{x}_i, \overline{x}_i$, and the regularizers are affected as well.

## 5 Experiments

**Models, Datasets and Hyper-parameters:** We evaluate the proposed regularizer on two datasets (MNIST and CIFAR10) with two different $\epsilon$ each. We consider only $\ell_\infty$ adversaries. We experiment with a variety of different network structures, including a MLP (2x100) with two 100-neuron hidden layers as (Salman et al., 2019), two Conv Nets (`Small` and `Large`) that are the same as (Wong et al., 2018), a family of 10 small conv nets and a family of 8 larger conv nets, all the same as (Zhang et al., 2019b), and also the same 5-layer convolutional network (`XLarge`) as in the latest version of CROWN-IBP (Zhang et al., 2019b). For CROWN-IBP, we use the updated expensive training schedule as (Zhang et al., 2019b), which uses 200 epochs with batch size 256 for MNIST and 3200 epochs with batch size 1024 for CIFAR10. We use up to 4 GTX 1080Ti or 2080Ti for all our experiments.

**Improved Training with Convex Relaxation** Table 1 shows comparisons with various approaches. All of our baseline implementations have improved compared with (Wong et al., 2018). When adding the proposed regularizers, the certified robust accuracy is further improved in all cases for both CP (Wong et al., 2018) and CROWN-IBP (Zhang et al., 2019b). We also provide results against

| Dataset | Model | Base Method | $\epsilon$ | $\lambda$ | $\gamma$ | Rob. Err | PGD Err | Std. Err |
|---|---|---|---|---|---|---|---|---|
| MNIST | 2x100, Exact | CP | 0.1 | 0 | 0 | 14.85% | **10.9%** | **3.65%** |
| MNIST | 2x100, Exact | CP | 0.1 | 2e-3 | 1 | **13.32%** | **10.9%** | 4.73% |
| MNIST | Small, Exact | CP | 0.1 | 0 | 0 | 4.47% | 2.4% | 1.19% |
| MNIST | Small, Exact | CP | 0.1 | 5e-3 | 5e-1 | **3.65%** | **2.2%** | 1.09% |
| MNIST | - | Best of PV[1] | 0.1 | - | - | 4.44% | 2.87% | 1.20% |
| MNIST | - | Best of RS[2] | 0.1 | - | - | 4.40% | 3.42% | **1.05%** |
| MNIST | Small | CP | 0.1 | 0 | 0 | 4.47% | **3.3%** | **1.19%** |
| MNIST | Small | CP | 0.1 | 0 | 5e-1 | **4.32%** | 3.4% | 1.51% |
| MNIST | Large | DAI[3] | 0.1 | - | - | 3.4% | 2.4% | 1.0% |
| MNIST | 2x100, Exact | CP | 0.3 | 0 | 0 | 61.39% | 49.4% | 33.16% |
| MNIST | 2x100, Exact | CP | 0.3 | 5e-3 | 5e-3 | **56.05%** | **44.3%** | **26.10%** |
| MNIST | Small, Exact | CP | 0.3 | 0 | 0 | 31.25% | 15.0% | 7.88% |
| MNIST | Small, Exact | CP | 0.3 | 5e-3 | 5e-1 | **29.65%** | **13.7%** | **7.28%** |
| MNIST | Small | CP | 0.3 | 0 | 0 | 42.7% | 26.0% | 15.93% |
| MNIST | Small | CP | 0.3 | 2e-3 | 2e-1 | **41.36%** | **24.0%** | **14.29%** |
| MNIST | XLarge | IBP[4] | 0.3* | - | - | 8.05% | 6.12% | **1.66%** |
| MNIST | XLarge | CROWN-IBP | 0.3* | - | - | 7.01% | 5.88% | 1.88% |
| MNIST | XLarge | CROWN-IBP | 0.3* | 0 | 5e-1 | **6.64%** | - | 1.76% |
| CIFAR10 | Small | CP | 2/255 | 0 | 0 | 53.19% | 48.0% | 38.19% |
| CIFAR10 | Small | CP | 2/255 | 5e-3 | 5e-1 | **51.52%** | **47.0%** | **37.30%** |
| CIFAR10 | Large | DAI[3] | 2/255 | - | - | 61.4% | 55.6% | 55.0% |
| CIFAR10 | XLarge | IBP[4] | 2/255* | - | - | 49.98% | 45.09% | 29.84% |
| CIFAR10 | XLarge | CROWN-IBP | 2/255* | - | - | 46.03% | 40.28% | 28.48% |
| CIFAR10 | Large | CP | 2/255 | 0 | 0 | 45.78% | **38.5%** | **29.42%** |
| CIFAR10 | Large | CP | 2/255 | 5e-3 | 5e-1 | **45.19%** | 38.8% | 29.76% |
| CIFAR10 | Small | CP | 8/255 | 0 | 0 | 75.45% | 68.3% | 62.79% |
| CIFAR10 | Small | CP | 8/255 | 1e-3 | 1e-1 | **74.70%** | **67.9%** | **62.50%** |
| CIFAR10 | Large | CP | 8/255 | 0 | 0 | 74.04% | 68.8% | **59.73%** |
| CIFAR10 | Large | CP | 8/255 | 5e-3 | 5e-1 | **73.74%** | **68.5%** | 59.82% |
| CIFAR10 | XLarge | IBP[4] | 8/255* | - | - | 67.96% | **65.23%** | **50.51%** |
| CIFAR10 | XLarge | CROWN-IBP | 8/255* | - | - | 66.94% | 65.42% | 54.02% |
| CIFAR10 | XLarge | CROWN-IBP | 8/255* | 0 | 5e-1 | **66.64%** | - | 53.78% |

Table 1: Results on MNIST, and CIFAR10 with small networks, large networks, and different coefficients of $d(x, \delta_0^*, W, b), r(x, \delta_0^*, W, b)$. All entries with positive $\lambda$ or $\gamma$ are using our regularizers. For all models not marked as "Exact", we have projected the input dimension of $\mathcal{W}_{i:1}$ to 50, the same as Wong et al. (2018). For $\epsilon$ values with *, larger $\epsilon$ is used for training. $\epsilon = 0.3, 2/255, 8/255$ correspond to using $\epsilon = 0.4, 2.2/255, 8.8/255$ for training respectively. For the methods: [1]: (Dvijotham et al., 2018a); [2]: (Xiao et al., 2018); [3]: (Mirman et al., 2018); [4] (Gowal et al., 2018).

a 100-step PGD adversary for our CP models. We will add PGD error rates for our CROWN-IBP models in our future version. Since both PGD errors and standard errors are reduced in most cases, the regularizer should have improved not only the certified upper bound, but also improved the actual robust error.

The relative improvement on 2x100 with our regularizer (10.3%/8.7%) are comparable to the improvements (5.9%/10.0%) from (Salman et al., 2019), despite the fact that we start from a stronger baseline. This indicates that the improvement brought by using our regularizer is comparable with using the expensive and unstable optimal layer-wise convex relaxation for relaxation.

Our results with Small are better than the best results of (Dvijotham et al., 2018a; Xiao et al., 2018) on MNIST with $\epsilon = 0.1$, though not as good as the best of (Mirman et al., 2018), which uses a larger model. When applying the same model on CIFAR10, we achieve better robust error than (Mirman et al., 2018).

The relative improvements in certified robust error for $\epsilon = 0.1$ and 0.3 are 18%/3.4% for the small exact model on MNIST, compared with 0.03%/3.13% for the random projection counterparts. In the exact models, we have better estimates of $\underline{x}_i, \overline{x}_i$. These consistent improvements validate that our proposed regularizers improve the performance.

In comparison with IBP-based methods, our regularizers is able to further improve CP on CIFAR10 with $\epsilon = 2/255$, and demonstrate the best result among all approaches compared in this setting. To our knowledge, this is the best result of models of the same size. By using our regularizers on CROWN-IBP to provide a better initialization for the later training stage of IBP, our method also achieves the best certified accuracy on CIFAR10 under $\epsilon = 8/255$. To provide more comprehensive evaluations, Table 4 shows the mean and variance of the results with smaller models, demonstrating consistent

| Dataset | $\epsilon(\ell_\infty)$ | Model Family | Method | Verified Test Error (%) | | | Standard Test Error (%) | | |
|---|---|---|---|---|---|---|---|---|---|
| | | | | best | median | worst | best | median | worst |
| MNIST | 0.3 | Gowal et al. (2018) | | 8.05 | - | - | 1.66 | - | - |
| | | 8 large models | CI Orig | 7.46 | 8.47 | 8.57 | 1.48 | 1.52 | 1.99 |
| | | | CI ReImp | 7.99 | 8.38 | 8.97 | 1.40 | 1.69 | 2.19 |
| | | | CI Reg | 7.26 | 8.44 | 8.88 | 1.51 | 1.72 | 2.21 |

Table 2: Our results on the MNIST dataset, with CROWN-IBP. CI Orig are results copied from the paper, CI ReImp are results of our implementation of CROWN-IBP, and CI Reg is with regularizer $r$.

improvements of our model, while Table 2 gives the best, median and worst case results with the large models on the MNIST dataset. Note all the networks are significantly smaller than (Gowal et al., 2018), and our batch size and number of epochs (at most 256 and 140) are also much smaller than (Gowal et al., 2018) (1600 and 3200). Still, we are able to achieve better results.

## 6 CONCLUSIONS

We propose two regularizers that lead to tighter LP relaxation bounds for certifiable robustness. Extensive experiments validate that the regularizers improve robust accuracy over non-regularized baselines. This work is a step towards closing the gap between certified and empirical robustness. Future directions include methods to improve computational efficiency for LP relaxations (and certified methods in general), and better ways to leverage random projections for acceleration.

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

## A  A TOY EXAMPLE FOR TIGHT RELAXATION

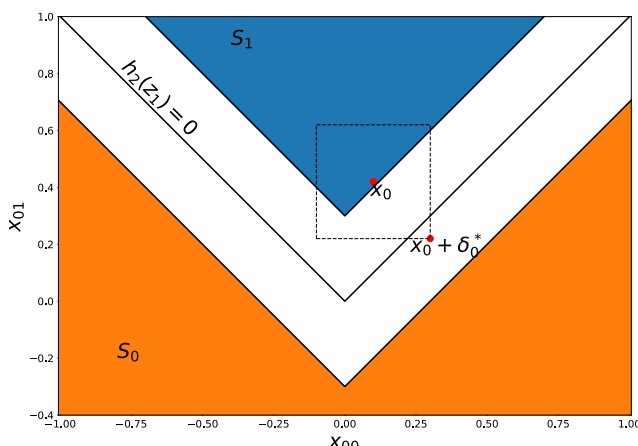

Figure 2: Illustration of the data distribution and the decision boundary of the network. In this case, $b = 0.3, \epsilon = 0.2, x_0 = [0.1, 0.42]^T$.

We give an illustrative example where the optimal solution to the relaxed problem is a feasible solution to the original non-convex problem for certain input samples even when unstable neurons exist. It is a binary classification problem for samples $x_0 = [x_{01}, x_{02}]^T \in \mathbb{R}^2$. We assume $x_0$ is uniformly distributed in $S_0 \cup S_1$, where $S_0 = \{x_0 | x_{02} \le |x_{01}| - b, \|x_0\|_\infty \le 1\}$, $S_1 = \{x_0 | x_{02} \ge |x_{01}| + b, \|x_0\|_\infty \le 1\}$, and $0 < b < 1$. The ground-truth label for $x_0 \in S_0$ and $x_0 \in S_1$ are 0, 1 respectively. The maximal-margin classifier for such data distribution is $\mathbb{1}_{\{z_{12} \ge |z_{12}|\}}$, where $z_1$ is the input to the classifier. The data distribution and the associated maximal-margin classifier is shown in Figure 2.

This maximal-margin classifier can be represented by a ReLU network with single hidden layer as $\mathbb{1}_{\{h_2(z_1) \ge 0\}}$, where $h_2(z_1) = W_2 \sigma(W_1 z_1)$, and

$$W_1 = \begin{bmatrix} 1 & 0 \\ -1 & 0 \\ 0 & 1 \\ 0 & -1 \end{bmatrix}, W_2 = \begin{bmatrix} -1 & -1 & 1 & -1 \end{bmatrix}. \tag{5}$$

**Claim 1** (Convex relaxation can be tight when unstable neurons exist). *The solution to the relaxed problem 15 is feasible for the original non-convex problem $\mathcal{O}$ of the aforementioned ReLU network $h_2(x_0 + \delta_0) = W_2 \sigma(W_1(x_0 + \delta_0))$ for any $x_0 \in S_1$ under any perturbation $\delta_0 \in \{\delta | \|\delta\|_\infty \le \epsilon, 0 < \epsilon < b\}$.*

*Proof.* Both the network and $S_1$ are symmetric in $x_{01}$, therefore it is sufficient to prove the result for $x_{01} \ge 0$.

**(1)** If $x_{01} \ge \epsilon$, since $x_{01} \in S_1$, $x_{02} \ge b + \epsilon$. The 4 neurons in $x_1 = W_1(x_0 + \delta_0)$ are either non-negative or non-positive for any $\delta_0 \in \{\delta | \|\delta\|_\infty \le \epsilon, 0 < \epsilon < b\}$ and the convex relaxation is tight.

**(2)** If $x_{01} < \epsilon$, for any input sample $x_0 \in \{x | x = [a, c]^T, 0 < a < b < b + a \le c\} \subseteq S_1$, the optimal perturbation $\delta_{\mathcal{O}}^* \in \{\delta | \|\delta\|_\infty \le \epsilon, 0 < a < \epsilon < b\}$ can be inferred from Figure 2 as $[\epsilon, -\epsilon]^T$. The corresponding ReLU activations and the optimal solution are

$$z_2' = [a + \epsilon, 0, c - \epsilon, 0]^T, x_2' = c - a - 2\epsilon. \tag{6}$$

Meanwhile, for the relaxed problem, the lower and upper bounds of the hidden neurons $x_1 = W_1 z_1$ are

$$\underline{x}_1 = [a - \epsilon, -a - \epsilon, c - \epsilon, -c - \epsilon]^T, \bar{x}_1 = [a + \epsilon, -a + \epsilon, c + \epsilon, -c + \epsilon]^T.$$

Therefore, the first 2 hidden neurons are unstable neurons, and the convex relaxation we are using will relax the ReLU operation $z_2 = \sigma(x_1)$ into $z_2 = D_1 x_1 + \delta_1$, where $D_1$ is a diagonal matrix, and $\delta_1$ are slack variables bounded by $0 \leq \delta_1 \leq \bar{\delta}_1$. The diagonal entries of $D_1$ and the upper bounds $\bar{\delta}_1$ are defined by Eq. 13 as

$$\mathrm{diag}(D_1) = \left[ \frac{a+\epsilon}{2\epsilon}, \frac{-a+\epsilon}{2\epsilon}, 1, 0 \right], \bar{\delta}_1 = \left[ \frac{(-a+\epsilon)(a+\epsilon)}{2\epsilon}, \frac{(-a+\epsilon)(a+\epsilon)}{2\epsilon}, 0, 0 \right]^T,$$

i.e., $\delta_{13}$ and $\delta_{14}$ are always 0. The relaxed linear network, as defined by the constraints in Eq. $\mathcal{C}$ with our specific relaxation, is now determined as $x_2 = h_2(z_1) = W_2(D_1 W_1(x + \delta_0) + \delta_1)$. It can be written into the same form as Eq. 14 as

$$c_t x_2 = c_t h_2(z_1) = c_t W_2 D_1 W_1(x + \delta_0) + c_t W_2 \delta_1 = c_t \mathcal{W}_{2:1} x + c_t \mathcal{W}_{2:1} \delta_0 + c_t W_2 \delta_1, \quad (7)$$

where

$$c_t = 1, \mathcal{W}_{2:1} = [-\frac{a}{\epsilon}, 1].$$

Therefore, to minimize the term $c_t \mathcal{W}_{2:1} \delta_0$, we should choose $\delta_0^* = [\epsilon, -\epsilon]^T$, which is equal to $\delta_{\mathcal{O}}^*$. To minimize $c_t W_2 \delta_1$, we should let $\delta_1^* = \left[ \frac{(-a+\epsilon)(a+\epsilon)}{2\epsilon}, \frac{(-a+\epsilon)(a+\epsilon)}{2\epsilon}, 0, 0 \right]$, which gives rise to the optimal solution of the relaxed problem as

$$z_2^* = D_1 W_1(x + \delta_0^*) + \delta_1^* = [a+\epsilon, 0, c-\epsilon, 0]^T, x_2^* = c - a - 2\epsilon,$$

the same as the optimal solution of the original non-convex problem given in Eq. 6. This shows both of the regularizers, in this case instantiated as

$$d(x, \delta_0^*, W_1, W_2) = c_t(x_2' - x_2^*), r(x, \delta_0^*, W_1, W_2) = \frac{1}{2} \left( |x_{11}' - x_{11}^*| + |x_{12}' - x_{12}^*| \right),$$

are able to reach 0 for certain networks and samples when non-stable neurons exist. $\square$

It might seem that adding $d(x, \delta_0^*, W, b) = p_{\mathcal{O}}'(x, \delta_0^*) - p_{\mathcal{C}}^*$ as a regularizer into the loss function will undesirably minimize the margin $p_{\mathcal{O}}'(x, \delta_0^*)$ for the ReLU network. Theoretically, however, it is not the case, since $\min p_{\mathcal{O}}' - p_{\mathcal{C}}^*$ is a different optimization problem from neither $\min p_{\mathcal{O}}'$ nor $\max p_{\mathcal{C}}^*$. In fact, the non-negative $d(x, \delta_0^*, W, b)$ could be minimized to 0 with both $p_{\mathcal{O}}'$ and $p_{\mathcal{C}}^*$ taking large values. In the illustrative example, it is easy to see that for any $x_0 \in \{x | x = [a, c]^T, 0 < a < b < b+a \leq c\}$, $d(x, \delta_0^*, W, b) = c_t(x_2' - x_2^*) = 0$, but $p_{\mathcal{O}}' = p_{\mathcal{C}}^* = c - a - 2\epsilon > 0$ when $\epsilon < \frac{b}{2}$.

Moreover, since we are maximizing $p_{\mathcal{C}}^*$ via the robust cross entropy loss[2] while minimizing the non-negative difference $p_{\mathcal{O}}' - p_{\mathcal{C}}^*$, the overall objective tends to converge to a state where both $p_{\mathcal{O}}'$ and $p_{\mathcal{C}}^*$ are large.

## B  PROOF OF PROPOSITION 1

**Proposition 1.** *Assume $\{z_i', x_i'\}_{i=1}^L$ is obtained by the ReLU network $h_L$ with input $z_1'$, and $\{\delta_i^*\}_{i=0}^{L-1}$ is the optimal solution of Fast-Lin or CROWN. If $z_1' = x + \delta_0^*$, and $x_{ij}' \in \mathcal{S}(\delta_{ij}^*)$ for all $i = 1, ..., L-1, j \in \mathcal{I}_i$, then $\{z_i', x_i'\}_{i=1}^L$ is an optimal solution of $\mathcal{O}$, Fast-Lin and CROWN. Here*

$$\mathcal{S}(\delta_{ij}^*) = \begin{cases} \{\underline{x}_{ij}, \overline{x}_{ij}\} & \text{if } \delta_{ij}^* = -\frac{\overline{x}_{ij}\underline{x}_{ij}}{\overline{x}_{ij} - \underline{x}_{ij}}, \\ \{0\} & \text{if } \delta_{ij}^* = 0. \end{cases}$$

*Proof.* We only need to prove $\{z_i', x_i'\}_{i=1}^L$ is an optimal solution of both Fast-Lin and CROWN. After that, $\{z_i', x_i'\}_{i=1}^L$ is both a lower bound and feasible solution of $\mathcal{O}$, and therefore is the optimal solution of $\mathcal{O}$.

Here we define $x_i^* = W_i z_i^* + b_i$ for $i = 1, ..., L$, $z_{i+1}^* = D_i^{(L)} x_i^* + \delta_i^*$ for $i = 1, ..., L-1$, and $z_1^* = x + \delta_0^*$. By definition, $\{x_i^*, z_i^*\}_{i=1}^L$ is an optimal solution of Fast-Lin or CROWN. Also, since

---

[2]The cross entropy loss on top of the lower bounds of margins to all non-ground-truth classes, see Eq. 7.

$z_1^* = z_1'$, we have $x_1' = W_1 z_1^* + b_1 = x_1^*$. Next, we will prove if the assumption holds, we will have $z_2' = z_2^*$ for both Fast-Lin and CROWN.

For $j \in \mathcal{I}_1^+$, by definition of Fast-Lin and CROWN, $D_{1j}^{(L)} = 1$ and $x_{1j}^* \geq \underline{x}_{1j} > 0$, so $x_{1j}' = x_{1j}^* \geq 0$, $z_{2,j}^* = D_{1j}^{(L)} x_{1j}^* = x_{1j}' = \max(x_{1j}', 0) = z_{2j}'$.

For $j \in \mathcal{I}_1^-$, again, by definition, $D_{1j}^{(L)} = 0$, and $x_{1j}^* \leq \bar{x}_{1j} < 0$, so $x_{1j}' = x_{1j}^* < 0$, $z_{2j}^* = D_{1j}^{(L)} x_{1j}^* = 0 = \max(x_{1j}', 0) = z_{2j}'$.

For $j \in \mathcal{I}_1$:

- If $\delta_{1j}^* = 0$ and $x_{1j}' = 0$ as assumed in the conditions, since $z_1^* = z_1'$, we know

$$x_{1j}^* = W_{1j} z_1^* + b_{1j} = W_{1j} z_1' + b_{1j} = x_{1j}' = 0,$$

where $W_{1j}$ is the $j$-th row of $W_1$. No matter what value $D_{1j}^{(L)}$ is, $z_{2,j}^* = D_{1j}^{(L)} x_{1j}^* = 0$, $z_{2j}' = \max(x_{1j}', 0) = 0$, the equality still holds.

- If $\delta_{1,j}^* = -\frac{\bar{x}_{1j}\underline{x}_{1j}}{\bar{x}_{1j}-\underline{x}_{1j}}$, for both Fast-Lin and CROWN, $z_{2j}^* = -\frac{\bar{x}_{1j}\underline{x}_{1j}}{\bar{x}_{1j}-\underline{x}_{1j}}(x_{1j}^* - \underline{x}_{1j})$. Further, if $x_{1j}^* \in \{\underline{x}_{1j}, \bar{x}_{1j}\}$ as assumed: if $x_{1j}^* = x_{1j}' = \underline{x}_{1j}$, then $x_{1j}' < 0$, so $z_{2j}' = \max(x_{1j}', 0) = 0$, and $z_{2j}^* = \frac{\bar{x}_{ij}}{\bar{x}_{ij}-\underline{x}_{ij}}(\bar{x}_{1j} - \underline{x}_{1j}) = 0 = z_{2j}'$; if $x_{1j}^* = x_{1j}' = \bar{x}_{1j}$, then $x_{1j}' > 0$, $z_{2j}' = x_{1j}'$ and $z_{2j}^* = \frac{\bar{x}_{ij}}{\bar{x}_{ij}-\underline{x}_{ij}}(\bar{x}_{1j} - \underline{x}_{1j}) = \bar{x}_{1,j} = z_{2j}'$.

Now we have proved $z_2' = z_2^*$ for both Fast-Lin and CROWN if the assumption is satisfied. Starting from this layer, using the same argument as above, we can prove $z_3' = z_3^*,...,z_{L-1}' = z_{L-1}^*$ for both Fast-Lin and CROWN. As a result, $x_L' = x_L^*$ and $c_t^T x_L' = c_t^T x_L^* = p_{\mathcal{C}}^*$, where $\mathcal{C}$ can be both Fast-lin and CROWN. Therefore, $\{z_i', x_i'\}_{i=1}^L$ is an optimal solution of $\mathcal{C}$. □

## C EXPERIMENTAL DETAILS

**Experiments for Table 1** The small convnet has two convolutional layers of 16, 32 output channels each and two FC layers with 100 hidden neurons. The large net has four Conv layers with 32, 32, 64 and 64 output channels each, plus three FC layers of 512 neurons. For all experiments, we are using Adam (Kingma & Ba, 2014) with a learning rate of $1e - 3$. Like (Wong et al., 2018), we train the models for 80 epochs, where in the first 20 epochs the learning rate is fixed but the $\epsilon$ increases from 0.01/0.001 to its maximum value for MNIST/CIFAR10, and in the following epochs, we reduce learning rate by half every 10 epochs. We do not use weight decay. We also adopt a warm-up schedule in all experiments, where $\lambda, \gamma$ increases form 0 to the preset values in the first 20 epochs, due to the noisy estimation from the random projections. Our implementation is based on the code released by (Wong et al., 2018), so when $\lambda = \gamma = 0$, we obtain the same results as as (Wong et al., 2018).

**Experiments for Other Tables** We use the same hyper-parameters as (Zhang et al., 2019b), except for "lr_decay_step" and "epochs", which are set to 20 and 140 for lower variance.

## D ABLATION STUDIES OF THE TWO REGULARIZERS

In this section, we give the detailed results with either $\lambda$ or $\gamma$ set to 0, i.e., we use only one regularizer in each experiment, in order to compare the effectiveness of the two regularizers. All the results are with the small model on CIFAR10 with $\epsilon = 2/255$. The best results are achieved with $r(x, \delta_0^*, W, b)$. We reasoned in 4.2 that $d(x, \delta_0^*, W, b)$ may not perform well when random projection is adopted. As shown in the supplementary, the best robust error achieved under the same setting when fixing $\gamma = 0$ is higher than when fixing $\lambda = 0$, which means $r(x, \delta_0^*, W, b)$ is more resistant to the noise introduced by random projection. Still, random projections offer a huge efficiency boost when they are used. How to improve the bounds while maintaining efficiency is an important future work.

| $\lambda$ | $\gamma$ | Robust Err (%) | Std. Err (%) | $\lambda$ | $\gamma$ | Robust Err (%) | Std. Err (%) |
|---|---|---|---|---|---|---|---|
| 1e-5 | 0 | 52.90 | 37.61 | 0 | 1e-3 | 52.56 | 37.45 |
| 5e-5 | 0 | 53.09 | 37.77 | 0 | 5e-3 | 53.38 | 38.19 |
| 1e-4 | 0 | 52.48 | 37.37 | 0 | 1e-2 | 52.60 | 37.73 |
| 5e-4 | 0 | 52.85 | 37.13 | 0 | 2.5e-2 | 52.70 | 37.78 |
| 1e-3 | 0 | 52.61 | 37.96 | 0 | 5e-2 | 53.13 | 38.36 |
| 2.5e-3 | 0 | 53.10 | 38.24 | 0 | 1e-1 | 52.72 | 38.22 |
| 5e-3 | 0 | 52.76 | 38.15 | 0 | 2.5e-1 | 52.90 | 38.04 |
| 1e-2 | 0 | 53.14 | 38.58 | 0 | 5e-1 | 52.39 | 37.48 |
| 5e-2 | 0 | 52.82 | 39.89 | 0 | 1 | **52.27** | **38.07** |
| 1e-1 | 0 | 53.94 | 41.59 | 0 | 2 | 53.10 | 38.64 |
| 5e-1 | 0 | 56.39 | 48.06 | | | | |
| 1 | 0 | 59.23 | 52.51 | | | | |

Table 3: Ablation results on CIFAR10 with the small model, where $\epsilon = 2/255$.

## E  DIFFICULTIES IN ADAPTING IBP FOR $\ell_2$ ADVERSARY

The Inverval Bound Propagation (IBP) method discussed here is defined in the same way as (Gowal et al., 2018), where the bound of the margins are computed layer-wise from the input layer to the final layer, and the bound of each neuron is considered independently for both bounding that neuron and using its inverval to bound other neurons.

It is natural to apply IBP against $\ell_\infty$ adversaries, since each neurons are allowed to change independently in its interval, which is similar to the $\ell_\infty$ ball. One way to generalize IBP to other $\ell_p$ norms is to modify the bound propagation in the first layer, such that any of its output neuron ($i$) is bounded by an interval centered at $x_{1i} = W_{1,i}x + b_{1,i}$ with a radius of $\epsilon_p \|W_{1,i}\|_{p^*}$, where $x_{1i}$ the clean image $x$'s response, and $W_{1,i}$ is the first layer's linear transform corresponding to the neuron, and $\|\cdot\|_{p^*}$ is the dual norm of $\|\cdot\|_p$, with $\frac{1}{p} + \frac{1}{p^*} = 1$. We refer to this approach IBP($\ell_p, \epsilon_p$). Here by the example of $\ell_2$ norm, we show such an adaptation may not be able to obtain a robust *convolutional* neural network compared with established results, such as reaching 61% certified accuracy on CIFAR10 with $\epsilon_2 = 0.25$ (Cohen et al., 2019).

Specifically, for adversaries within the $\ell_2$-ball $\mathcal{B}_{2,\epsilon_2}(x)$, IBP($\ell_2, \epsilon_2$) computes the upper and lower bounds as

$$\bar{x}_{1i}^2 = W_{1,i}x + \epsilon_2\|W_{1,i}\|_2 + b_{1,i}, \ \underline{x}_{1i}^2 = W_{1,i}x - \epsilon_2\|W_{1,i}\|_2 + b_{1,i}. \tag{8}$$

By comparison, for some adversary within the $\ell_\infty$-ball $\mathcal{B}_{\infty,\epsilon_\infty}(x)$, IBP($\ell_\infty, \epsilon_\infty$) computes the upper and lower bounds as

$$\bar{x}_{1i}^\infty = W_{1,i}x + \epsilon_\infty\|W_{1,i}\|_1 + b_{1,i}, \ \underline{x}_{1i}^\infty = W_{1,i}x - \epsilon_\infty\|W_{1,i}\|_1 + b_{1,i}. \tag{9}$$

Since the two approaches are identical in the following layers, to analyze the best-case results of IBP($\ell_2, \epsilon_2$) based on established results of IBP($\ell_\infty, \epsilon_\infty$), it suffices to compare the results of IBP($\ell_\infty, \epsilon_\infty$) with $\epsilon_\infty$ set to some value such that the range $\bar{x}_1^\infty - \underline{x}_1^\infty$ of Eq. 9 is majorized by the range $\bar{x}_1^2 - \underline{x}_1^2$ of Eq. 8. In this way, we are assuming a weaker adversary for IBP($\ell_\infty, \epsilon_\infty$) than the original IBP($\ell_2, \epsilon_2$), so its certified accuracy is an upper bound of IBP($\ell_2, \epsilon_2$). Therefore, it suffices to let

$$\epsilon_\infty\|W_{1,i}\|_1 \leq \epsilon_2\|W_{1,i}\|_2, \ \forall i = 1, 2, ..., n_1. \tag{10}$$

For any $W_{1,i} \in \mathbb{R}^d$, we have $\|W_{1,i}\|_1 \leq \sqrt{d}\|W_{1,i}\|_2$. To make Eq. 10 hold for any $W_{1,i} \in \mathbb{R}^d$, we can set

$$\epsilon_\infty = \frac{1}{\sqrt{d}}\epsilon_2.$$

In general, $d$ is equal to the data dimension, such as 3072 for the CIFAR10 dataset. However, for convolutional neural networks, the first layer is usually convolutional layers and $W_{1,i}$ is a 3072-dimensional sparse vector with at most $k \times k \times 3$ non-zero entries at fixed positions for convolution kernels with size $k$ and input images with 3 channels. In (Zhang et al., 2019b; Gowal et al., 2018; Wong et al., 2018), $k = 3$ for their major results. In this case,

$$\epsilon_\infty \geq \frac{1}{3\sqrt{3}}\epsilon_2. \tag{11}$$

Under such assumptions, for $\epsilon_2 = 0.25$, the certified accuracy of IBP($\ell_2$, 0.25) on CIFAR10 should be upper bounded by IBP($\ell_\infty$, 0.04811), unless changing the first layer bounds into $\ell_\infty$ norm based bounds significantly harms the performance.[3] The best available results of certified accuracies are 33.06% for IBP($\ell_\infty$, 0.03137) and 23.20% for IBP($\ell_\infty$, 0.06275) (Zhang et al., 2019b). Comparing with the established results from (Cohen et al., 2019) (61%), we can conclude the certified accuracy of IBP($\ell_2$, 0.25) is at least 27.93% to 37.80% lower than the best available results, since we are assuming a weaker adversary.

IBP($\ell_2$, $\epsilon_2$) is also not as good as the results with convex relaxation from (Wong et al., 2018), where the best single-model (with projection as approximation) certified accuracy with $\epsilon_2 = 36/255$ is 51.09%. For IBP, this adversary is no weaker than $\epsilon_\infty = 6.9282/255$. The best available results for IBP($\ell_\infty$, 2/255) and IBP($\ell_\infty$, 8/255) are 50.02% (Gowal et al., 2018) and 33.06% (Zhang et al., 2019b) respectively, which indicates the certified accuracy of IBP($\ell_2$, 36/255) is at least 1.07% to 18.03% worse (much loser to 18.03%) than the approximated version of convex relaxation under the same $\ell_2$ adversary.

## F    SOLUTIONS TO THE RELAXED PROBLEMS

In this section, we give more details about the optimal solutions of Fast-Lin (Weng et al., 2018) and CROWN (Zhang et al., 2018), to make this paper self-contained. Recall that for layer-wise convex relaxations, each neuron in the activation layer are independent. $\{\underline{a}_{ij}, \underline{b}_{ij}, \overline{a}_{ij}, \overline{b}_{ij}\}$ are chosen to bound the activations assuming the lower bound $\underline{x}_{ij}$ and upper bound $\overline{x}_{ij}$ of the preactivation $x_{ij}$ is known. For $j \in \mathcal{I}_i$, $\underline{a}_{ij} x_{ij} + \underline{b}_{ij} \leq \max(0, x_{ij}) \leq \frac{\overline{x}_{ij}}{\overline{x}_{ij} - \underline{x}_{ij}}(x_{ij} - \underline{x}_{ij}) \leq \overline{a}_{ij} x_{ij} + \overline{b}_{ij}$; for $j \in \mathcal{I}_i^+$, $\underline{a}_{ij} = \overline{a}_{ij} = \overline{a}_{ij} = \overline{b}_{ij} = 1$; for $j \in \mathcal{I}_i^-$, $\underline{a}_{ij} = \overline{a}_{ij} = \underline{b}_{ij} = \overline{b}_{ij} = 0$.

**Optimal Solutions of Fast-Lin**    In Fast-Lin (Weng et al., 2018), for $j \in \mathcal{I}_i$, $\underline{a}_{ij} = \overline{a}_{ij} = \frac{\overline{x}_{ij}}{\overline{x}_{ij} - \underline{x}_{ij}}$, $\underline{b}_{ij} = 0$, $\overline{b}_{ij} = -\frac{\overline{x}_{ij} \underline{x}_{ij}}{\overline{x}_{ij} - \underline{x}_{ij}}$, shown as the blue region in the right of Figure 1. To compute the lower and upper bound $\underline{x}_{ij}$ and $\overline{x}_{ij}$ for $x_{ij}$, we just need to replace the objective of Eq. $\mathcal{C}$ with $c_{ij}^\top x_i$, where $c_{ij}$ is a one-hot vector with the same number of entries as $x_i$ and the $j$-th entry being 1 for $\underline{x}_{ij}$ and -1 for $\overline{x}_{ij}$ (an extra negation is applied to the minimum to get $\overline{x}_{ij}$).

Such constraints allow each intermediate ReLU activation to reach their upper or lower bounds independently. As a result, each intermediate *unstable neuron* can be seen as an adversary adding a perturbation $\delta_{ij}$ in the range $[0, -\frac{\overline{x}_{ij} \underline{x}_{ij}}{\overline{x}_{ij} - \underline{x}_{ij}}]$ to a linear transform, represented as $z_{ij} = \frac{\overline{x}_{ij}}{\overline{x}_{ij} - \underline{x}_{ij}} x_{ij} + \delta_{ij}$. Such a point of view gives rise to a more interpretable explanation for Fast-Lin. If we construct a network from the relaxed constraints, then the problem becomes how to choose the perturbations for both the input and intermediate unstable neurons to minimize $c_t^\top x_L$ of a multi-layer linear network. Such a linear network under the perturbations is defined as

$$z_{i+1} = D_i x_i + \delta_i, x_i = W_i z_i + b_i, \text{ for } i = 1, ..., L, z_1 = x + \delta_0, \tag{12}$$

where $D_i$ is a diagonal matrix and $\delta_i$ is a vector. The input perturbation satisfies $\|\delta_0\|_p \leq \epsilon$. The $j$-th diagonal entry $D_{ij}$ and the $j$-th entry $\delta_{ij}$ for $i > 0$ is defined as

$$D_{ij} = \begin{cases} 0, & \text{if } j \in \mathcal{I}_i^- \\ 1, & \text{if } j \in \mathcal{I}_i^+, \\ \frac{\overline{x}_{ij}}{\overline{x}_{ij} - \underline{x}_{ij}}, & \text{if } j \in \mathcal{I}_i \end{cases} \delta_{ij} \in \mathcal{S}_{\delta_{ij}} = \begin{cases} \left[0, -\frac{\overline{x}_{ij} \underline{x}_{ij}}{\overline{x}_{ij} - \underline{x}_{ij}}\right], & \text{if } j \in \mathcal{I}_i \\ \{0\}, & \text{otherwise.} \end{cases} \tag{13}$$

With such an observation, we can further unfold the objective in Eq. $\mathcal{C}$ 12 into a more interpretable form as

$$c_t^\top x_L = c_t^\top f_L(D_{L-1} f_{L-1}(\cdots D_1 f_1(x))) + c_t^\top \sum_{i=0}^{L-1} W_L \prod_{k=i+1}^{L-1} D_k W_k \delta_i, \tag{14}$$

where the first term of RHS is a forward pass of the clean image $x$ through a linear network interleaving between a linear layer $x = W_i z + b_i$ and a scaling layer $z = D_i x$, and the second term

---

[3]Which is unlikely, since $\epsilon_\infty$ is now a weaker adversary than $\epsilon_2$ and the difference in gradient expression only appears in the first layer.

is the sum of the $i$-th perturbation passing through all the weight matrices $W_i$ of the linear operation layers and scaling layers $D_i$ after it.

Therefore, under such a relaxation, only the second term is affected by the variables $\{\delta_i\}_{i=0}^{L-1}$ for optimizing Eq. $\mathcal{C}$. Denote the linear network up to the $i$-th layer as $g_i(x)$, and $\mathcal{W}_{i:i'} = W_i \prod_{k=i'}^{i-1} D_k W_k$. We can transform Eq. $\mathcal{C}$ with $\underline{a}_{ij} = \overline{a}_{ij} = \frac{\overline{x}_{ij}}{\overline{x}_{ij} - \underline{x}_{ij}}$, $\underline{b}_{ij} = 0$, $\overline{b}_{ij} = -\frac{\overline{x}_{ij}\underline{x}_{ij}}{\overline{x}_{ij} - \underline{x}_{ij}}$ into the following constrained optimization problem

$$\underset{\delta_0, \ldots, \delta_{L-1}}{\text{minimize}} \; c_t^\top g_L(x) + \sum_{i=0}^{L-1} c_t^\top \mathcal{W}_{L:i+1} \delta_i, \text{ subject to } \|\delta_0\|_p \leq \epsilon, \delta_{ij} \in \mathcal{S}_{\delta_{ij}} \text{ for } i = 1, \ldots, L-1. \quad (15)$$

Notice $c_t^\top \mathcal{W}_{L:i+1}$ is just a row vector. For $i > 0, j \in \mathcal{I}_i$, to minimize $c_t^\top \mathcal{W}_{L:i+1} \delta_{ij}$, we let $\delta_{ij}$ be its minimum value 0 if $(c_t^\top \mathcal{W}_{L:i+1})_j \geq 0$, or its maximum value $\delta_{ij} = -\frac{\overline{x}_{ij}\underline{x}_{ij}}{\overline{x}_{ij} - \underline{x}_{ij}}$ if $(c_t^\top \mathcal{W}_{L:i+1})_j < 0$, where $(c_t^\top \mathcal{W}_{L:i+1})_j$ is the $j$-th entry of $c_t^\top \mathcal{W}_{L:i+1}$. For $\delta_0$, when the infinity norm is used, we set $\delta_{ij} = -\epsilon$ if $(c_t^\top \mathcal{W}_{L:i+1})_j \geq 0$, and otherwise $\delta_{ij} = \epsilon$. For other norms, it is also easy to see that

$$\min_{\|\delta_0\|_p \leq \epsilon} c_t^\top \mathcal{W}_{L:1} \delta_0 = -\epsilon \max_{\|\delta_0\|_p \leq 1} -c_t^\top \mathcal{W}_{L:1} \delta_0 = -\epsilon \|c_t^\top \mathcal{W}_{L:1}\|_*, \quad (16)$$

where $\|\cdot\|_*$ is the dual norm of the $p$ norm. In this way, the optimal value $p_{\mathcal{C}}^*$ $p_{\mathcal{C}}$ of the relaxed problem (Eq. 15) can be found efficiently without any gradient step. The optimal value can be achieved by just treating the input perturbations and intermediate relaxed ReLU activations as adversaries against a linear network after them. The resulting expression for the lower-bound is

$$p_{\mathcal{O}}^* \geq p_{\mathcal{C}}^* = c_t^\top g_L(x) - \epsilon \|c_t^\top \mathcal{W}_{L:1}\|_* - \sum_{i=1}^{L-1} \sum_{j \in I_i} \frac{\overline{x}_{ij}\underline{x}_{ij}}{\overline{x}_{ij} - \underline{x}_{ij}} \min((c_t^\top \mathcal{W}_{L:i+1})_j, 0). \quad (17)$$

Though starting from different points of view, it can be easily proved that the objective derived from a dual view in (Wong et al., 2018) is the same as Fast-Lin.

**Optimal Solution of CROWN** The only difference between CROWN and Fast-Lin is in the choice of $\underline{a}_{ij}$ for $j \in \mathcal{I}_i$. For ReLU activations, CROWN chooses $\underline{a}_{ij} = 1$ if $\overline{x}_{ij} \geq -\underline{x}_{ij}$, or $\underline{a}_{ij} = 0$ otherwise. This makes the relaxation tighter than Fast-Lin, but also introduces extra complexity due to the varying $D_i$. In Fast-Lin, $D_i$ is a constant once the upper and lower bounds $\overline{x}_i$ and $\underline{x}_i$ are given. For CROWN, since $0 < \overline{a}_{ij} < 1$, $\overline{a}_{ij} \neq \underline{a}_{ij}$, $D_i$ now changes with the optimality condition of $\delta_i$, which depends on the layer $l$ and the index $k$ of the neuron/logit of interest. Specifically, for $\ell_\infty$ adversaries, the optimality condition of $\delta_i$ is determined by $c_{lk}^\top \mathcal{W}_{l:i}$, so now we have to apply extra index to the slope as $D_i^{(l,k)}$, as well as the equivalent linear operator as $\mathcal{W}_{l:1}^{(l,k)}$. As a result, the optimal solution is now

$$c_{lk}^\top g_l(x) - \epsilon \|c_{lk}^\top \mathcal{W}_{l:1}^{(l,k)}\|_* - \sum_{i=1}^{L-1} \sum_{j \in I_i} \frac{\overline{x}_{ij}\underline{x}_{ij}}{\overline{x}_{ij} - \underline{x}_{ij}} \min((c_{lk}^\top \mathcal{W}_{l:i+1}^{(l,k)})_j, 0). \quad (18)$$

This drastically increase the number of computations, especially when computing the intermediate bounds $\underline{x}_i$ and $\overline{x}_i$, where we can no longer just compute a single $\mathcal{W}_{l:1}$ to get the bound, but have to compute number-of-neuron copies of it for the different values of $D_i^{(l,k)}$ in the intermediate layers.

**Practical Implementations of the Bounds** In practice, the final output bound (also the intermediate bounds) is computed in a backward pass, since we need to determine the value $(c_{lk}^\top \mathcal{W}_{l:i+1}^{(l,k)})_j$ to choose the optimal $\delta_{ij}^*$, which is the multiplication of all linear operators after layer $i$. Computing $c_{lk}^\top \mathcal{W}_{l:i+1}^{(l,k)}$ in a backward pass avoids repeated computation. It proceeds as

$$c_{lk}^\top \mathcal{W}_{l:i}^{(l,k)} = c_{lk}^\top \mathcal{W}_{l:i+1}^{(l,k)} D_i^{(l,k)} W_i. \quad (19)$$

## G    DETAILED RESULTS ON THE FAMILY OF SMALL MODELS

| Method | error | model A | model B | model C | model D | model E | model F | model G | model H | model I | model J |
|---|---|---|---|---|---|---|---|---|---|---|---|
| Copied | std. (%) | $5.97 \pm .08$ | $3.20 \pm 0$ | $6.78 \pm .1$ | $3.70 \pm .1$ | $3.85 \pm .2$ | $3.10 \pm .1$ | $4.20 \pm .3$ | $2.85 \pm .05$ | $3.67 \pm .08$ | $2.35 \pm .09$ |
| | verified (%) | $15.4 \pm .08$ | $10.6 \pm .06$ | $16.1 \pm .3$ | $11.3 \pm .1$ | $11.7 \pm .2$ | $9.96 \pm .09$ | $12.2 \pm .6$ | $9.90 \pm .2$ | $11.2 \pm .09$ | $9.21 \pm .3$ |
| Baseline | std. (%) | $5.65 \pm .04$ | $3.23 \pm .3$ | $4.70 \pm .4$ | $2.94 \pm .05$ | $6.39 \pm .3$ | $2.89 \pm .05$ | $4.11 \pm .3$ | $2.55 \pm .1$ | $3.30 \pm .4$ | $2.56 \pm .1$ |
| | verified (%) | $14.70 \pm .07$ | $10.65 \pm .3$ | $13.78 \pm 1.$ | $9.89 \pm .3$ | $15.26 \pm .6$ | $9.54 \pm .1$ | $12.06 \pm .9$ | $8.92 \pm .2$ | $10.81 \pm .8$ | $9.67 \pm .4$ |
| With $r$ | std. (%) | $5.80 \pm .04$ | $3.16 \pm .06$ | $5.16 \pm .3$ | $3.06 \pm .05$ | $6.15 \pm .2$ | $2.95 \pm .05$ | $3.83 \pm .2$ | $2.57 \pm .1$ | $3.29 \pm .04$ | $2.73 \pm .4$ |
| | verified (%) | $14.54 \pm .07$ | $10.46 \pm .08$ | $13.37 \pm .3$ | $9.91 \pm .3$ | $14.43 \pm .3$ | $9.48 \pm .2$ | $11.01 \pm .5$ | $8.83 \pm .3$ | $10.18 \pm .5$ | $9.49 \pm .4$ |

Table 4: Mean and standard deviation of the family of 10small models on MNIST with $\epsilon = 0.3$. Baseline is CROWN-IBP with epoch=140 and lr_decay_step=20. Like in CROWN-IBP, we run each model 5 times to compute the mean and standard deviation. "Copied" are results from (Zhang et al., 2019b).

