# OpenReview forum: "Improved Training of Certifiably Robust Models"
_ICLR.cc/2020/Conference — Reject_

### Official Review · AnonReviewer2 · 2019-10-21
**Official Blind Review #2**

**Rating:** 6

**Review:**

Strengths:
This work proposed two regularizers that can be used to train neural networks that yield convex relaxations with tighter bounds.
The experiments display that the proposed regularizations result in tighter certification bounds than non-regularized baselines.
The problem is interesting, and this work seems to be useful for many NLP pair-wise works.
weaknesses:
Some presentation issues.
The dataset, MNIST, is not good enough for a serious research.
More datasets need to be added to the experiments in this paper.


Comments:
This paper proposes two regularizers to train neural networks that yield convex relaxations with tighter bounds.

Overall, the paper solves an interesting problem. Though I did not check complete technical details, the extensive evaluation results seem promising.

1. There are some presentation issues that can be addressed. For example, on page 8, the sentence of “the family of 10small” misses a blank space.

2. In the experiments, the dataset is not a good one for evaluating the performance of the proposed idea.

In conclusion,  at this stage, my opinion on this paper is Weak Accept.

**Experience Assessment:**

I have published one or two papers in this area.

**Review Assessment: Checking Correctness Of Derivations And Theory:**

I carefully checked the derivations and theory.

**Review Assessment: Checking Correctness Of Experiments:**

I carefully checked the experiments.

**Review Assessment: Thoroughness In Paper Reading:**

I read the paper thoroughly.

---

> ### Author Response · Authors · 2019-11-15
> **Thank you for your feedback!**
>
> Thank you for you acknowledgement of our work! We have made a major revision to improve our presentation, and hope you still love this version! In fact, we already had experimental results on CIFAR10 in our previous version. In this revision, we have added more results on CIFAR10 comparing with stronger baselines such as IBP and CROWN-IBP. We are able to achieve the best certified accuracy under both $\epsilon=2/255$ and $\epsilon=8/255$. While we have not expanded the experiments to even more datasets, we will try our best to do so in the future.

---

### Official Review · AnonReviewer3 · 2019-10-26
**Official Blind Review #3**

**Rating:** 3

**Review:**

Summary:
The paper proposes two new regularizers for adversarial robustness inspired by literature on verification of ReLU neural networks for resilience to epsilon perturbations using convex relaxations. The paper shows empirically that the proposed method leads to better robustness than previous works.

Strengths:
+ The paper seems to have an interesting perspective (with the proposed looser relaxation) of the convex relaxation of an adversary adding noise at every layer in the network

Weaknesses:

*Sec. 4.1: Eqn. (O) does not have a convex relaxation, it is the exact problem which is intractable. Why are we comparing the optimal values of p*(O) and p*(C)? The paper from Salman et.al. already shows that there is a convex relaxation barrier, which essentially corresponds to this difference. In general, in Sec. 4, it is often unclear whether when we talk about p(O) if we are referring to the unrelaxed original problem or the tightest convex relaxation. For example, at the start of Sec. 4.1, it seems like we are talking about the convex relaxation and then in Sec. 4.3 it seems like we are talking about the unrelaxed problem.

*It is not clear how/ why the proposed method of relaxing (which by the way seems identical to Fast-Lin (Weng et.al.) is better than the optimal convex relaxation. Would this not lead to looser bounds? Is that the thing we are looking to investigate? Making that more clear would be useful. Perhaps it would be good to argue the proposed regularizer in this work cannot be constructed with the optimal convex relaxation. Is that true? A discussion on this would be helpful.

* The crux of the contribution seems to rest on the premise that identifying the optimal perturbation in the input space with the relaxed model, and then computing the activations with respect to that and forcing the forward pass to saturate near the margins of the relu polytope (relaxation) is a good idea. In general, it seems very unclear why this should work based on the evidence presented in the paper. Specifically with the relaxation, it might not even be guaranteed (as far as I understand) that the value of \delta_0^* that is found from problem C is even going to lie inside the L\inf norm ball around the point x, for example. Thus it is not clear to me if this is an approach for verification or a regularizer based on verification.

* Ultimately, the value of the approach in this context (as per my understanding) comes from the experiments and the results which show that there is increased robustness. It would be great to clarify a couple of details in the experiments:
1. Is the method of Wong et.al. using the looser convex relaxation (used here) or the tight convex relaxation when reporting the numbers in Table. 1?
2. If the optimal convex relaxation can be used to construct the same regularizer as the one proposed here, it would be good to evaluate how well that does.

Overall, I am not an expert in the area but a lot of details from the writing (such as point 1 under weakness) and the theoretical justification of the regularizer are unclear to me. Thus given these (perceived) weaknesses I would lean towards weak rejection. Clarifications on these points would help me revise my score.

**Experience Assessment:**

I do not know much about this area.

**Review Assessment: Checking Correctness Of Derivations And Theory:**

I did not assess the derivations or theory.

**Review Assessment: Checking Correctness Of Experiments:**

I assessed the sensibility of the experiments.

**Review Assessment: Thoroughness In Paper Reading:**

I read the paper at least twice and used my best judgement in assessing the paper.

---

> ### Author Response · Authors · 2019-11-15
> **Thank you for your feedback! [1]**
>
> Thank you for your acknowledgement of our work and the valuable feedback! We have revised our paper and hope you love the current version. Below we try our best to address your specific concerns.
>
> > "-  *Sec. 4.1: Eqn. (O) does not have a convex relaxation, it is the exact problem which is intractable. Why are we comparing the optimal values of p*(O) and p*(C)? … In general, in Sec. 4, it is often unclear whether when we talk about p(O) if we are referring to the unrelaxed original problem or the tightest convex relaxation…"
> We are sorry for the typos and have corrected it in the latest version. Throughout the paper we are trying to minimize the gap between the optimal values of the original non-convex problem $\mathcal{O}$ and its convex relaxation $\mathcal{C}$ on each individual training sample. We have shown with the new illustrative example in Appendix A that the gap between $p^*_{\mathcal{O}}$ and $p^*_{\mathcal{C}}$ can be 0 for a large portion of samples, and with empirical results on practical networks and datasets, we have shown that minimizing such gap improves robustness.
>
> > "-  It is not clear how/ why the proposed method of relaxing (which by the way seems identical to Fast-Lin (Weng et.al.) is better than the optimal convex relaxation. Would this not lead to looser bounds? Is that the thing we are looking to investigate? .... Perhaps it would be good to argue the proposed regularizer in this work cannot be constructed with the optimal convex relaxation. Is that true? A discussion on this would be helpful."
>
> We have revised Proposition 1 to prove that when the same condition holds, i.e., $r=0$, both CROWN and Fast-Lin are tight. In the same way, the optimal layer-wise convex relaxation will also be tight. Still, we would like to make some clarifications here:
> 1) We are not proposing a new convex relaxation method in the paper and try to beat the optimal layer-wise convex relaxation (referring it to LP-All). Instead, we propose two regularizers based on over observations from Fast-Lin that can be used on top of established convex relaxation bounds to train certifiably robust ReLU networks, and demonstrate an improved certified accuracy. The regularizers can be easily extended to different forms of convex relaxations, including CROWN and CROWN-IBP, and we have demonstrated the improvements in our experiments.
> 2) Theoretically, since the feasible set of LP-All is a subset of Fast-Lin, LP-All is tighter than Fast-Lin. However, the benefit of our regularizer lies in obtaining a model that can be better certified by the looser Fast-Lin, not from introducing a theoretically tighter bound. If we use LP-All to verify the models obtained with our regularizers, we could get even better results.
> 3) More specifically, in our experiments, we compare the improvements in certified robust accuracy by either (1) train the model (a small 2-hidden-layer MLP) using our regularizers and using Fast-Lin to verify, or (2) verifying the model trained with Wong et al. (equivalent to Fast-Lin) by using LP-All instead of Fast-Lin (numbers taken from Salman et al. (2019)). Our regularizers results in networks that can be better certified by the looser Fast-Lin, and the improvement is comparable to using the much more expensive LP-All to provide a better bound for models trained without our regularizers.
> 4) For LP-All, such regularizers can also be applied, but since LP-All is an impractical approach for training robust networks, it is hard for us to provide any experimental results. Without developing anything new, the first regularizer can be applied directly to LP-All by minimizing the difference between the lower bounds of margin given by LP-All and the margins obtained by plug in the “optimal” perturbation.
> 5) For the second regularizer, one can still compute it from the Fast-Lin solutions and minimize it. If the conditions in Proposition 1 is satisfied for the Fast-Lin bounds, the gap will also vanish for LP-All, since the three points (three red dots in the right of Figure 1 in the revised version) are also feasible for LP-All. Further, one can identify the optimal solutions for the unstable neurons $x_{ij}^*$ of LP-All, and try to minimize the distance between $x’_{ij}$ computed by the ReLU network when the input is $x+\delta_0^*$, but it is unclear whether such a process is easily differentiable (w.r.t. the parameters of the network).

---

> ### Author Response · Authors · 2019-11-15
> **Thank you for your feedback! [2]**
>
> > "- * The crux of the contribution seems to rest on the premise that identifying the optimal perturbation in the input space with the relaxed model, … In general, it seems very unclear why this should work based on the evidence presented in the paper. Specifically with the relaxation, it might not even be guaranteed … that the value of \delta_0^* that is found from problem C is even going to lie inside the L\inf norm ball around the point x, for example. Thus it is not clear to me if this is an approach for verification or a regularizer based on verification."
> First, it is guaranteed that the value of $\delta_0^*$ found from problem $\mathcal{C}$ satisfies the norm ball constraint, since this constraint is contained in Eq. ($\mathcal{C}$) and we are finding the optimal solution for $\mathcal{C}$ that satisfies all its constraints.
> Our approach is for training a network that has better verified robustness. It is a regularization based on Fast-Lin, which also improves the tightness of CROWN-IBP as well as other convex relaxations for ReLU networks. We aim to train a neural network that can be better verified by these bounds. Meanwhile, but enforcing a tighter bound during training, it can also alleviate the problem of over regularization caused by loose bounds in theory. It enforces the convex relaxation to be tight for samples from the data distribution, and improves the verified accuracy on test set empirically.
>
>
> > "- 1. Is the method of Wong et.al. using the looser convex relaxation (used here) or the tight convex relaxation when reporting the numbers in Table. 1? "
> All the results in Table 1 are verified with Fast-Lin or equivalently Wong et al.’s bound. The optimal layer-wise convex relaxation is hardly applicable to the networks in Table 1.
>
> > "-  2. If the optimal convex relaxation can be used to construct the same regularizer as the one proposed here, it would be good to evaluate how well that does."
> In theory, for the optimal layer-wise convex relaxation, our regularizers can be applied, and only the second regularizer needs to be adapted to allow $x_{ij}’$ to lie on the line of ReLU constraint (left of Figure 1) when $\delta_{ij}^*=0$. However, as we have mentioned before, this relaxation is too expensive to be integrated into the training process. See the experimental results in the CROWN paper (Efficient Neural Network Robustness Certification with General Activation Functions), where LP-Full as referred to in the paper is orders of magnitude more expensive than CROWN, and often fail to converge in a reasonable amount of time. Even CROWN is already expensive enough.

---

### Official Review · AnonReviewer4 · 2019-10-28
**Official Blind Review #4**

**Rating:** 3

**Review:**

Summary:
The aim of the paper is to improve verified training. One of the problem with verified training is the looseness of the bounds employed so the authors suggest incorporating a measure of that looseness into the training loss. It is based on a reformulation of the relaxation of Weng et al.


Comments:
Page 2: "it can certify a broader class of adversaries that IBP cannot certify, like the `2 adversaries.". You can definitely use IBP to very properties against L2-adversaries. It is simply a matter of changing the way the bound is propagated through the first layer.
Page 3: It's a bit pedantic, but the convex relation of Ehlers (middle of figure 1) is not the optimal convex relaxation. It is optimal only if you assume that all ReLU non linearities are relaxed independently. See the work by Anderson et al. for some examples
Page 5, section 4: "We investigate the gap between the optimal convex relaxationin Eq. O" There is a bit of confusion in this section. Eq O is not the optimal convex relaxation, it's the hard non-convex problem.
Section 4.1 bothers me. Equation C is the relaxed version of equation O, so they are only going to be equal if there is essentially no relaxation going on. Saying that it's possible to check whether the equivalence exists is a bit weird. The only case where this can happen is if all the terms in the sum over I_i are zero, which is essentially going to mean that no ReLU is ambiguous. (or if the c W are all positives, but that would be problematic during the optimization of the intermediate bounds given that c would make them both signs then)
Page 5, section 4.2: The authors suggest minimizing d, the gap between the value of the bound obtained, and the value of forwarding the solution of the relaxation through the actual network. Essentially, this would amount to maximizing the lower bound (which all verified training already does), at the same time as minimizing the value of the margin (p_O) on a point of the neighborhood for which we want robustness (x + delta_0). Minimizing the value of the margin is the opposite of what we would want to do, so I'm not surprised by the observation of the author that this doesn't work well.
The conclusion of the section that d can not be optimized to 0 also seems quite obvious if you think about what the problem is.

Section 4.3:
"the optimal solution of C can only be on the boundary of the feasible set." -> There is a subtlety here that I think the authors don't address. The three points they identify are the only feasible optimal solutions for solving a linear program over the feasible domain given by the relaxation of one ReLU but, when solving over the whole of C, the solution needs to be on the boundary of the feasible domain of C, which is larger than those three points.

The whole section is quite convoluted and makes very strong assumption. For Proposition 1, the condition x \in S(\delta) means that all the intermediate bound in the network must have been tight (so that the actual forwarding of an x can match the upper or lower bound used in the relaxation), and that the optimal solution of the relaxation requires all intermediate points to be at either at their maximum or their minimum. The only case I can visualise for this is essentially once again the case where there are no ambiguous ReLU and the full thing is linear.

Regarding the experiments section, it would be benefical to include in table 1 the results of Gowal et al. (On the Effectiveness of Interval Bound propagation for Training Verifiably Robust Models) for better context. The paper is already cited so it should have been possible to include those numbers, which are often better than the ones reported here.
The comparison is included in table 2, when the baseline is beaten, but this is with using the training method of CROWN-IBP and it seems like most of the improvements is due to CROWN-IBP.

Typos/minor details:
Page 2: " In addition, a bound based on semi-definite programming (SDP) relaxation was developed and minimized as the objective Raghunathan et al. (2018). (Wong & Kolter, 2017) presents an upper bound" -> citation format
Page 8: "CORWN-IBP "

Opinion:
I think that the analysis section is pretty confusing and needs to be re-thought. It provides a lot of complex discussion of when the relaxation will be exact, without really identifying that it will be when you have very few ambiguous ReLU. I think that there might be a few parallels to identify between the regularizer proposed and the ReLU stability one of Xiao et al. (ICLR2019) from that aspect. The experimental results are not entirely convicing due to the lack of certain baselines.

**Experience Assessment:**

I have published in this field for several years.

**Review Assessment: Checking Correctness Of Derivations And Theory:**

I carefully checked the derivations and theory.

**Review Assessment: Checking Correctness Of Experiments:**

I assessed the sensibility of the experiments.

**Review Assessment: Thoroughness In Paper Reading:**

I read the paper thoroughly.

---

> ### Author Response · Authors · 2019-11-15
> **Thank you for your detailed feedback! [1]**
>
> Thank you for your valuable time and the interesting discussions! We have made major revisions to our paper and hope you like this new version. We have added two sections into the appendix to address your concerns about killing the ambiguous neurons and the $\ell_2$ IBP, and added more experimental results against stronger baselines into Table 1.
>
> > "- You can definitely use IBP to verify properties against L2-adversaries. It is simply a matter of changing the way the bound is propagated through the first layer."
> Indeed, you are correct: IBP can definitely certify other adversaries by modifying the first-layer propagation. We have corrected the sentence in our paper. However, given the established results, it seems that IBP cannot do it well in some important cases if we only modify the way the bound is propagated through the first layer, such as training robust convolutional networks against l2 adversaries. We have added some discussions in Appendix E to demonstrate that this approach cannot lead to better results (often much worse according to our estimation) than established results. We show that the certified accuracy of this first-layer-$\ell_2$ IBP is at least 27.93 to 37.80 lower than established results of randomized smoothing at $\epsilon_2$=0.25, and at least 18.03 lower (approximately) than the results of the approximated convex relaxation at $\epsilon_2$=36/255 by Wong et al. (2018). We hope Appendix E addresses your concerns, but please let us know if we misunderstood your idea about adapting IBP to other adversaries.
>
>  > "- the convex relation of Ehlers (middle of figure 1) is not the optimal convex relaxation. It is optimal only if you assume that all ReLU non linearities are relaxed independently."
> Thank you for pointing this out. We have revised the captions of Figure 1 and other related contents to address this point. Yes, same as in (Salman et al. 2019), the optimal convex relaxation refers to the optimal convex relaxation of the nonlinear constraint $z_{i+1}=\sigma(x_i)$ from each single layer, which is just like the middle of Figure 1 for unstable neurons in ReLU networks. In our previous version, we referenced it as “the optimal relaxation for each $j\in \mathcal{I}_i$”, which is a bit vague but can still be correct if understood as the optimal relaxation for the single constraint given by  $j\in \mathcal{I}_i$ if it is considered independently.
>
> > "- Eq O is not the optimal convex relaxation, it's the hard non-convex problem."
> Sorry for the confusion. The typo has been corrected. What we really want to say is to investigate the gap between Eq. O and Eq. C.
>
> > "- Equation C is the relaxed version of equation O, so they are only going to be equal if there is essentially no relaxation going on. … The only case where this can happen is if all the terms in the sum over $I_i$ are zero, which is essentially going to mean that no ReLU is ambiguous. "
> Notice for both Eq. C and Eq. O, the input $x$ is a given constant. We are only enforcing the relaxed solution to be equal to the non-convex solution at the training samples, instead of letting these two solutions to be equivalent in the whole input space. There exist specific networks and (a significant portion of) samples where solutions to Eq. C and Eq. O are equivalent but ambiguous ReLUs still exist (see the illustrative example in Claim 1 of Appendix A), and by using such a regularizer we expect the bound to be tight for samples from the data distribution.
>
> > "- Page 5, section 4.2: The authors suggest minimizing d …  this would amount to maximizing the lower bound (which all verified training already does), at the same time as minimizing the value of the margin (p_O) on a point of the neighborhood for which we want robustness (x + delta_0). Minimizing the value of the margin is the opposite of what we would want to do, so I'm not surprised by the observation of the author that this doesn't work well. "
> By “doesn’t work well”, we just want to say it does not work as well as the second regularizer when used separately; the first regularizer still improves the results upon the baseline. See the results from both Table 1 and Table 3 for the small model with $\epsilon=2/255$ on CIFAR10.
> Moreover, minimizing $d$, i.e., $\min (p’_{\mathcal{O}}-p^*_{\mathcal{C}})$, is definitely different from both minimizing $p’_{\mathcal{O}}$ and maximizing $p^*_{\mathcal{C}}$ as you suggested, since the solution to the first optimization problem could be the case where both $p’_{\mathcal{O}}$ and $p^*_{\mathcal{C}}$ are large but the difference $(p’_{\mathcal{O}}-p^*_{\mathcal{C}})$ is small.
> In fact, since we are minimizing the robust cross entropy loss which pushes $p^*_{\mathcal{C}}$ to larger values while minimizing this gap $d$, the model tends to converge to a state where both $p^*_{\mathcal{C}}$ and $p’_{\mathcal{O}}$ are large.
> We have also included a discussion based on the illustrative example at the end of Appendix A.

---

> > ### Author Response · Authors · 2019-11-15
> > **Thank you for your feedback! [2]**
> >
> > > "- ...The three points they identify are the only feasible optimal solutions for solving a linear program over the feasible domain given by the relaxation of one ReLU but, when solving over the whole of C, the solution needs to be on the boundary of the feasible domain of C, which is larger than those three points."
> > Thank you for the careful check. We were actually referring to the layer-wise convex relaxation, Fast-Lin, instead of any convex relaxation.  We have made an edit and the current version emphasizes the unstable neurons are independent. For Fast-Lin, the solution to each unstable neuron can be considered independently, and the intersection of their feasible set with the original ReLU constraint are the three points. It is sufficient to consider the three points for each unstable neuron to check whether the relaxed solution of Fast-Lin is feasible for the non-convex problem. It is also a sufficient condition for CROWN to be tight. We have added this conclusion in Proposition 1.
> >
> > > "-  For Proposition 1, the condition x \in S(\delta) means that all the intermediate bound in the network must have been tight (so that the actual forwarding of an x can match the upper or lower bound used in the relaxation), and that the optimal solution of the relaxation requires all intermediate points to be at either at their maximum or their minimum. The only case I can visualise for this is essentially once again the case where there are no ambiguous ReLU and the full thing is linear.."
> > This assumption is not strong at all for certain samples and networks, as can be seen from Appendix A, where the conditions in Proposition 1 can be satisfied for a large portion of the data distribution even when unstable neurons exist and the network is nonlinear in the norm ball. Also, we are only matching the optimal solution of $\mathcal{C}$ inside the norm ball with the corresponding solution in $\mathcal{O}$, not enforcing equivalence in the whole norm ball. So and the ReLU network does not have to be linear inside the whole norm ball.
> >
> > > "-  Regarding the experiments section, it would be benefical to include in table 1 the results of Gowal et al…. for better context. "
> > We have applied our regularizer to CROWN-IBP, which now achieves better results than both the CROWN-IBP baseline and IBP (Gowal et al.). The results has been added into Table 1.
> > Our method is orthogonal to CROWN-IBP, and we believe further improvements upon CROWN-IBP is valuable.
> > In fact, convex relaxation is better than CROWN-IBP in some cases where $\epsilon$ is small, such as when $\epsilon=2/255$ on CIFAR10. We have added the new comparisons into Table 1.
> >
> > > "-  I think that the analysis section is pretty confusing and needs to be re-thought. It provides a lot of complex discussion of when the relaxation will be exact, without really identifying that it will be when you have very few ambiguous ReLU. I think that there might be a few parallels to identify between the regularizer proposed and the ReLU stability one of Xiao et al. (ICLR2019) from that aspect. The experimental results are not entirely convicing due to the lack of certain baselines."
> > As we have demonstrated in Appendix A, the existence of ambiguous ReLU does not eliminate the possibility that the bound is tight; the bound could still be tight even when ambiguous ReLU exists for certain input samples. Reducing the number of ambiguous ReLUs could improve tightness if we assume the range for other ambiguous ReLUs does not increase, but it is not the other way round, i.e., tightness at the sample distribution does not necessarily mean the elimination of ambiguous ReLU. It only requires the optimal solutions to match at a certain adversarial perturbation, not inside the whole norm ball. Therefore, in theory, our regularizer does not explicitly kill the ambiguous ReLUs, which is different from Xiao et al. (ICLR2019).
> > We have also added comparisons against stronger baseline methods (IBP, CROWN-IBP) into Table 1.

---

### Author Response · Authors · 2019-11-15
**Sorry for the late response!**

Dear reviewers,

We are sorry for our late response! We were trying our best to come up with a better version of the paper. In this revision, we have changed the structure of writing, moved some discussions into the appendix. Also, to address your concerns about the mechanism of our regularizer, we have also added an illustrative example into the appendix to show that our regularizer does not necessarily kill the unstable neurons. We have also added new experimental results, and now achieve state-of-the-art certified accuracies on both MNIST and CIFAR10. We hope you love this version!

---

### Decision · Program_Chairs · 2019-12-19

**Decision:**

Reject

**Comment:**

The authors develop regularization schemes that aim to promote tightness of convex relaxations used to provides certificates of robustness to adversarial examples in neural networks.

While the paper make some interesting contributions, the reviewers had several concerns on the paper:
1) The aim of the authors' work and the distinction with closely related prior work is not clear from the presentation. In particular, the relationship to the ReLU stability regularizer (Xiao et al ICLR 2019) and the FastLin/CROWN-IBP work (https://arxiv.org/abs/1906.06316) is not very well presented in the theoretical sections or the experiments.

2) The theoretical results (proposition 1) requires very strong conditions to apply, which are unlikely to be satisfied for real networks. This calls into question the effectiveness of the framework developed by the authors.

While the paper has some interesting ideas, it seems unfit for publication in its present form.